# Packham’s Triumph Pears (*Pyrus communis* L.) Post-Harvest Treatment during Cold Storage Based on Chitosan and Rue Essential Oil

**DOI:** 10.3390/molecules26030725

**Published:** 2021-01-30

**Authors:** Yeimmy Peralta-Ruiz, Carlos David Grande-Tovar, Diana Paola Navia Porras, Angie Sinning-Mangonez, Johannes Delgado-Ospina, María González-Locarno, Yarley Maza Pautt, Clemencia Chaves-López

**Affiliations:** 1Faculty of Bioscience and Technology for Food, Agriculture and Environment, University of Teramo, Via R. Balzarini 1, 64100 Teramo, Italy; yyperaltaruiz@unite.it (Y.P.-R.); jdelgado1@usbcali.edu.co (J.D.-O.); cchaveslopez@unite.it (C.C.-L.); 2Facultad de Ingeniería, Programa de Ingeniería Agroindustrial, Universidad del Atlántico, Carrera 30 Número 8–49, 081008 Puerto Colombia, Colombia; alsinning@mail.uniatlantico.edu.co; 3Grupo de Investigación de fotoquímica y fotobiología, Universidad del Atlántico, Carrera 30 Número 8–49, 081008 Puerto, Colombia; mclaudiagonzalez@mail.uniatlantico.edu.co (M.G.-L.); ymaza@mail.uniatlantico.edu.co (Y.M.P.); 4Grupo de Investigación Biotecnología, Facultad de Ingeniería, Universidad de San Buenaventura Cali, Carrera 122 # 6–65, 76001 Cali, Colombia; dpnavia@usbcali.edu.co

**Keywords:** antifungal, chitosan, *Ruta graveolens* essential oil

## Abstract

Pears (*Pyrus communis* L.) cv. Packham’s Triumph are very traditional for human consumption, but pear is a highly perishable climacteric fruit with a short shelf-life affected by several diseases with a microbial origin. In this study, a protective effect on the quality properties of pears was evidenced after the surface application of chitosan-*Ruta graveolens* essential oil coatings (CS + RGEO) in four different concentrations (0, 0.5, 1.0 and 1.5 %, *v*/*v*) during 21 days of storage under 18 °C. After 21 days of treatment, a weight loss reduction of 10% (from 40.2 ± 5.3 to 20.3 ± 3.9) compared to the uncoated pears was evident with CS + RGEO 0.5%. All the fruits’ physical-chemical properties evidenced a protective effect of the coatings. The maturity index increased for all the treatments. However, the pears with CS + RGEO 1.5% were lower (70.21) than the uncoated fruits (98.96). The loss of firmness for the uncoated samples was higher compared to the coated samples. The pears’ most excellent mechanical resistance was obtained with CS + RGEO 0.5% after 21 days of storage, both for compression resistance (7.42 kPa) and force (22.7 N). Microbiological studies demonstrated the protective power of the coatings. Aerobic mesophilic bacteria and molds were significantly reduced (in 3 Log CFU/g compared to control) using 15 µL/mL of RGEO, without affecting consumer perception. The results presented in this study showed that CS + RGEO coatings are promising in the post-harvest treatment of pears.

## 1. Introduction

Pear (*Pyrus communis* L.) is one fruit with high production figures in Latin America, with around one million tons annually [1]. China is the major producer in Asia with about 56 million tons, while Italy does the same in Europe, followed by the United States, Spain, and Turkey [2]. The immense worldwide production of pears also brings significant losses, with farmers and producers as the main affected supply chain sectors. Pear shelf-life depends on several factors such as growing season climate, nutrition, environmental conditions, cultivar, and microbial attacks, causing losses in the pear industry ranging from 5% to 50% of production [2,3]. The primary post-harvest diseases in pear, such as blue mold, pink mold, grey mold, and black spot, are related mainly to fungi of the genus *Mucor piriformis, Penicillium expansum,* and *Botrytis cinerea* [4].

Pear fruits are climacteric, and coating is considered one of the most popular techniques to prolong shelf life [5]. Several studies have evaluated the effect of edible coatings containing polysaccharides, such as soy protein isolate (SPI), in combination with hydroxypropyl methylcellulose (HPMC) and olive oil on ‘Babughosha’ Pears (*Pyrus communis* L.) stored at ambient temperature (28 ± 5 °C and 60 ± 10% Relative Humidity (RH)). Various edible coatings such as zein-oleic acid [6], plant oils [7], Shellac, Semperfresh, and carboxymethylcellulose [8] are useful on pears.

Chitosan (CS), a deacetylated derivative of chitin, is deeply investigated for many industrial and medical applications [9]. Chitosan coatings demonstrated an improvement in the storability of several perishable fruits. Barrier properties against moisture and oxygen and a capacity to preserve volatile bioactive compounds as well as a potent antimicrobial and food quality preservation capacity have been observed [10]. Some authors have evaluated chitosan coatings on pears obtaining a significantly delayed ripening and improved fruit firmness [11,12]. Nonetheless, chitosan coatings present a disadvantage as their hydrophilic nature forces the introduction of hydrophobic composites such as some essential oils, which also provide antioxidant and antimicrobial properties to the food during the post-harvest stage [13].

On the other hand, the treatment of crops with essential oils (EOs) could be a safer and alternative option for post-harvest preservation. Essential oils (EOs) are complex mixtures of terpenes and terpenoids, coumarins, and homologs of phenylpropanoids, with biological properties and sometimes strong odors. *Ruta graveolens* plant has been identified as a source of different antifungal compounds such as 5- and 8-methoxy psoralen, with potent activity against fungi such as *Rhizoctonia solanii*, *Fusarium* spp., *Pyrenochaeta lycopersici*, *Trichoderma viride*, and *Penicillium* spp. [14]. In previous studies, *Ruta graveolens* essential oil (RGEO) has shown high activity against the fungi *Aspergillus fumigatus*, *Cladosporium herbarum*, *Candida albicans*, and *Colletotrichum gloeosporioides* [15,16]. However, despite their antimicrobial activity, EOs are volatile and could influence the food’s organoleptic properties with the possible presence of phytotoxicity [17]. A more recent and useful strategy uses composites of biopolymers and essential oils as coatings on the fruit surfaces for a controlled release of the volatile compounds, inhibiting microbial growth on the fruit surface and conserving more time nutritional composition, quality, and product acceptance [18].

Despite all the information above, no literature using Chitosan (CS) + RGEO coatings to preserve pears’ post-harvest decay is available. Taking into account the antifungal activity of RGEO components and the efficiency of CS + RGEO in maintaining guava [19], cape gooseberries [20], papaya [16], and tomato [21] quality properties while inhibiting microbial growth, the objective of this study was to evaluate the effect of the application of CS + RGEO coatings on Packham’s Triumph pears to preserve their quality during cold storage.

## 2. Results and Discussion

Data regarding the characterization by mass spectrometry-gas chromatography (MS-GC) of the RGEO and characterization of the CS + RGEO emulsions were reported in our previous work (Appendix A) [19]. The emulsions’ pH was slightly acidic in all cases (around 4.42 to 4.45) for chitosan dissolution. At the same time, the density, apparent viscosity, and particle size values of the FFE demonstrated significant reductions (*p* < 0.05) with the introduction of the RGEO (Appendix A).

### 2.1. Physical-Chemical Analysis of Fruits after CS + RGEO Coating

#### 2.1.1. pH Analysis

None of the pH formulations present significant differences concerning the control or between them, except for days 10 and 21 (Figure 1). On these days, the formulations added a protective effect and retained the organic acid content in the fruit; however, no effect is seen from the inclusion of the RGEO to the CS formulation. The statistical analysis for each formulation relating to storage days showed no significant difference between the first day and the final day for the formulations, confirming the coatings’ protective trend. Uncoated fruits presented significant differences concerning the factor days caused by fast organic acid consumption. The present study results agree with previous reports showing similar trends in pears with chitosan-based coatings stored at 20 °C with a pH ranging between 4.2 and 4.6 [22]. The ripening processes demand high energy from different carbon compounds (e.g., organic acids, amino acids, and sugars) in the metabolism pathways [23]. The pH usually increases during the ripening of a climacteric fruit due to the organic acid consumption for the metabolic processes during fruit respiration [24]. It is noteworthy that some authors suggested a reduction in the fruit’s rate metabolism [25] as a phenomenon caused by a barrier effect of the coatings—behavior that may explain the lower pH values in coated pears.

#### 2.1.2. Titratable Acidity (TA)

Titratable acidity (TA) can be correlated with the primary organic acids in fruits, mainly citric acid and maleic acid in pears, according to the proteomic analysis in the ripening process [26]. The formulations do not show significant differences between them or the control for days 0, 3, 7, 10, and 14 (Figure 2). On day 21, coated pears with F2, F3, and F5 tended to maintain significantly higher TA levels than the control, demonstrating protection (lower consumption of organic acid and ripening) during the final day of the storage time. The highest level of TA (0.17%) was recorded in pears coated with F5. As in the case of pH, no effect is observed of the different concentrations of RGEO in formulations with CS. The TA was also significantly different (*p* < 0.05) on day 21 compared to the first day with the uncoated pears and F4. The other formulations do not present significant differences (*p* < 0.05) with the storage time, confirming the protective effect of F2, F3, and F5. Similar trends have been observed with pears coated with CS-based coatings, corroborated by Rosenbloom et al. [12] and Lin et al. [27]. The change in TA with storage time in uncoated fruits may be due to the normal ripening process [28], which is slowed down with the formulations of CS and CS + RGEO by the barrier effect of the coatings against oxygen, which inhibits the oxidation of the organic acids [29]. Previous studies have demonstrated that chitosan-based coatings reduce citric and malic acid contents, which are the major organic acids in ripe pear fruit [26]. Organic acids can support numerous and diverse functions in plants, associated with the supply of carbon skeletons during the plant growth process or the critical role of malate in photosynthesis and stomatal regulation [30].

#### 2.1.3. Soluble Solids Content (SS)

To better characterize the CS-RGEO coating effects on pears, we analyzed the soluble solids content. The results shown in Figure 3 indicated statistically significant differences (*p* < 0.05) in the SS content between F5 and F4, F5 regarding uncoated pears for days 10 and 21. No significant effects in the soluble solids total were evidenced between the pears coated with the formulations, indicating there is no more significant influence of the coatings with the inclusion of RGEO. However, the lowest percentage of soluble solids (11.45%) on the last day of storage was recorded in pears coated with CS + RGEO 1.5% (F5).

The uncoated pears increased significantly from 10.3% of SS to 12.8% after 21 days of storage at 18 °C, while pears coated with F5 did not present significant differences and had a minor increase in all formulations with a change from 10.4% to 11.6%, demonstrating a preservation effect previously seen with the pH and titratable acidity. These results were consistent with the result showed by Rosenbloom et al. [12] with CS-based coatings. It is observed that coated fruits have a minor change in the soluble solids during storage time. According to the maturation stage and stress conditions, the ripening process drastically alters the biochemistry and physiology of climacteric fruits such as pears due to by-products and changes produced in the secondary metabolism. Sugar accumulation (mainly glucose, fructose, and sucrose) occurs exponentially upon the cell division phase to finally plateau at the end of the ripening process, while organic acid contents substantially decrease [31]. Soluble solids content also increases with starch degradation due to the increased hydrolase activities and ATP metabolic energy content [32].

#### 2.1.4. Maturity Index

The maturity index is the ratio between the percentage of soluble solids and the percentage of acidity. Figure 4 shows that all the samples increased the maturation index during the storage time, but slower than the control. All the formulations kept a significantly lower maturation index than the uncoated samples (F1) for the 21st day. Despite the formulations not presenting significant differences between them, lower values for the maturity index were obtained with F3 (RGEO 0.5%) and F5 (RGEO 1.5%) with 73 and 71%, respectively. These results showed the same trend, followed by properties already studied where all the formulations have a protective effect on the pears. The formulations presented significant differences regarding the storage time, with an increase since day 7. However, the change in the value of the maturity index was lower in the coated fruits. The results obtained in this study are in agreement with previous studies in guava [19], cape gooseberries [20], tomatoes [21], and papaya [16] coated with CS + RGEO. The main modifications during fruit ripening are changes in color and texture based on sugar, organic acid, and volatile compounds, contributing to the balance between sugar and organic acids [26]. The changes in the percentage of soluble solids and the percentage of acidity and, therefore, in the maturity index during the storage of fruits are the ripening process results [25]. On the other hand, an increase in the soluble solids level during storage is because starch and nutrients can degrade into sugars and soluble substances [33]. The chemical composition of the RGEO, mainly composed of ketones and sesquiterpenes with the ability to modify the internal atmosphere of the fruits, can also contribute to the protective effect of CS-based coatings [19].

#### 2.1.5. Weight Loss

Weight loss percentage is an essential indicator of the degree of preservation of fruits and vegetables. As observed in Table 1, F2 showed no significant differences with the control on no day of the study. F5 and F3, F4, F5 showed significantly (*p* < 0.05) lower weight losses than the control for day 14 and 21, respectively, according to the Duncan test. On day 21, there are no significant differences between the CS and CS + RGEO formulations. However, a protective effect of the RGEO is evident with weight losses less than the control in a range between 41 and 50%. All the samples showed significant (*p* < 0.05) weight losses with time according to the LSD Fischer test by day 21 concerning day 3. Several authors have also reported the same trend of a reduction in weight loss rate in fruits with CS-based coatings reinforced with essential oils. Peralta et al. [16] reported weight losses of 20% in papayas coated with CS-RGEO formulations after 12 days of storage, evidencing a good behavior of the coatings in this study. Our results in this study evidence good protective behavior of the coatings with similar values but more extended storage. Cosme Silva et al. [34] suggested that the CS-based coatings reduce the fruits’ respiratory rate, acting as a barrier and decreasing water loss and solutes. Usually, the weight loss corresponds to the unbonded water released to the environment [35]. Our results agree with those reported by various authors, who suggest that the inclusion of the hydrophobic compounds in the CS formulations provides a better water loss barrier [13]. The RGEO reinforced the coatings and decreased weight losses in the pears.

#### 2.1.6. CO_2_ Respiration Rate

Pears are stored under low oxygen conditions that rely on the hypoxic metabolism pathway inside the fruit. The high resistance to pears’ gas transport due to the highly dense tissues produces a high respiration rate associated with ripening with local anoxia during controlled atmosphere storage [36]. Decreasing respiratory O_2_ consumption in response to a reduction in the available O_2_ has been described in different plants and tissues [37]. While respiratory metabolism is likely to be affected by local anoxia due to limited gas transport, respiratory metabolism’s active regulation may also play a role [38]. Figure 5 represents the CO_2_ respiration rate values, expressed as mg of CO_2_ kg^−1^h^−1^, on days 0, 3, 7, 10, 14, and 21.

Figure 5 shows the high respiration rate of uncoated fruits at the beginning of the study compared to coated fruits presenting significant differences. After three and seven days, there was an apparent decrease in the respiration rate due to the treatments. On day 14, no significant differences were observed between treatments. On the last day, F3 had a significant decrease concerning the control and the other formulations. In addition, it showed significant differences regarding factor time on day 21 compared to day 0. F4 and F5 had a higher respiration rate among the treatments.

Chitosan and essential oil coatings have been previously documented as suitable treatments with the capacity to inhibit O_2_ consumption and CO_2_ production, generating lesser gas permeability during the first ten storage days. After that, the barrier effect is lost as a result of the deterioration of the coatings. Coating hydration and oil evaporation could explain the decreasing barrier effect, and the fruit will continue in its climacteric peak due to senescence [19].

As stated before, a modification in the oxygen atmosphere would modify respiration rates, decreasing oxygen consumption and stimulating an anoxic fermentative metabolism [39]. A balance between oxygen and anoxic respiration pathways must avoid excessive fermentation, which negatively affects the sensorial attributes of fruits [40]. Simultaneously, the accumulation of high CO_2_ concentration could also harm fruit quality by accelerating color and firmness changes and increasing pectic compounds’ solubilization [41]. However, reducing the respiration rate generates less ethylene for maturation oxygen consumption, reduces respiration, and extends the fruit’s shelf life. After day ten, it is possible to say that the respiration rates were low, but uncoated fruits had lower respiration than those with coatings, probably due to coating degradation. A sensory panel was consulted to control that adverse changes in sensorial quality aspects were not presented. The barrier effect generated by the coatings of CS + RGEO did not allow the rising of strange flavors or aromas. Different chitosan-essential oil coatings have demonstrated a reduction in respiration rates in apples [42], strawberries [43], and table grapes [44].

#### 2.1.7. Mechanical Properties

Softening or loss of firmness in fruits increases due to metabolic processes fostered by depolymerization of the cell wall components over time, exacerbated by increased respiration of the fruit [45]. Figure 6 shows an increase in the percentage of deformation and a decrease in both compressive Strength (kPa) and force (N) with increasing days of storage. Regarding compression resistance (Figure 6a), on day 14, F3 presents significant differences concerning the control. For the other days, the formulations do not present significant differences between them or with the control. In addition, no significant differences were found between the force (N) (Figure 6b). Likewise, it is observed that concerning deformation (Figure 6c) did not present significant differences between formulations or with the control from day 3. Despite not having statistically significant differences in the mechanical properties of the fruits coated with the different formulations, the texture properties of the pears were conserved for all treatments. Similar results were reported by Dai et al. [46], who evaluated coatings based on starch and starch nanocrystals on pears. In this study, even though there were no significant differences between the treatments, they did occur between coated and uncoated pears.

The formulation that provides the most mechanical resistance to the pear is CS + RGEO 0.5% (F3), which presents the highest value after 21 days of storage, both for compression resistance (7.1 kPa) and force (20.1 N). That the coating with less RGEO presented better mechanical property values may be related to saturation. This phenomenon indicates a limiting concentration of essential oil, above which a decrease in mechanical properties is promoted; the excess increases intermolecular mobility, favoring the solubilization of components and forming different bonds that could reduce the coating’s cohesion, preventing it from fulfilling its barrier function [47]. F3 presented a lower value of weight loss on day 21.

The preceding suggests that the treatment with more excellent resistance to compression or greater firmness on the last day of storage manages to maintain mechanical resistance due to the inhibition of water loss closely related to weight loss. Similar results were reported previously, where chitosan-essential oil coatings reduced transpiration, increased water retention, provided turgor to the fruit cells, and preserved firmness [48]. In addition, CS + RGEO retains fruit firmness by reducing water loss and cell wall degradation because of the mold-forming fungi inhibition on the fruit’s infected surfaces [16].

#### 2.1.8. Color Parameters

The color changes in fruits are related to the ripening process, but it is also the fruit’s response to some alterations caused by diseases or improper post-harvest handling. It is an important parameter that must be controlled because it is the first cause of consumer rejection if the consumer is not satisfied. Figure 7 shows the color changes in the L, a*, and b* coordinates of the pears subjected to the treatments over time. When adding the coatings on day zero, no significant differences were observed between the treatments and the control for L and b*—only a present significant difference for a* but with a minimal change, which indicates that the coatings did not alter the original color of the pears, presenting the transparency expected of an edible coating. Peralta et al. [16] also reported no significant change in color parameters between uncoated papaya and coated fruits.

From day 14, significant differences (*p* < 0.05) attributed to the pears’ ripening process and the appearance of peel physiopathies [49] that become evident with ripening, such as senescent breakdown, were observed. On day 21, the L* parameter was significantly decreased in the samples that did not contain RGEO in the coating (Control and CS treatments, F1 and F2) compared to the samples that contained RGEO (Figure 7a). The L* parameter’s decrease is related to moisture loss in the fruit peel [50] and non-enzymatic browning reactions [51]. This indicates that CS + RGEO at the concentrations of the study reduces the permeability of the fruit skin, which is possible due to the covalent interactions between the network and the components of the essential oil, reducing the hydrophilic groups available to form hydrophilic bonds in the matrix (CS + RGEO) with water [52].

The parameter a* increased from day 14, indicating the loss of pears’ green color related to its ripening process (Figure 7b). It was evidenced that the control sample increased significantly compared with the other treatments, presenting little differences between the different concentrations of RGEO, indicating a relationship with the respiration rate decreasing, caused mainly by the polymer matrix chitosan [53]. The parameter b* significantly decreased (*p* < 0.05) from day 10 to the control, indicating the loss of the yellow color of pears (Figure 7c). This is associated with the oxidation of polyphenolic compounds released from the vacuoles, oxidized on the surface due to the air’s oxygen (senescent breakdown) and chlorophylls’ degradation [49]. The samples with RGEO significantly decreased b* concerning the uncoated pears on day 21. This result suggests that the observed maturation process’ decrease may be related to protecting the polyphenolic compounds present on the pear’s surface by the coating [54]. Similar results were previously obtained in other studies with chitosan-essential oil coatings in papaya [16], guava [19], and strawberries [55], where the changes in color parameters were delayed in coated fruits compared to uncoated fruit.

### 2.2. Decay Evaluation

#### 2.2.1. Severity Index (SI)

It is well known that at the post-harvest stage, *Penicillium expansum*, *Botrytis cinerea*, and *Mucor piriformis* are the primary fungi affecting pears, contributing to the SI due to the microbial infection and physical damage produced [4]. The SI is a visual parameter to measure the physical damage evolution of fruits during storage, indicating the skin’s external damage. All the formulations presented a significant decrease in the SI compared to uncoated pears (Figure 8). On day 21, significant differences between CS + RGEO and the CS formulation were observed. F4 and F5 (CS + RGEO 1.0% and CS + RGEO 1.5%) preserved the pears until day 21 without any physical or microbiological damage. Uncoated samples ranged from 0 on day 0 to 4.0 on day 21 of the storage. The formulation containing CS + RGEO 0.5% presented a protective effect, with only 0.7% of SI on day 21. Formulation F2 also presented a protective effect, with 1.7% of SI on day 21. Figure 8 shows that only the control had significant differences for SI according to the LSD Fisher Test regarding factor time, evidencing the positive effect of the formulations on the conservation of pears. These results demonstrate that CS presented a protective effect against physical and microbiological damage, acting as a positively reinforced barrier with the addition of RGEO in a concentration-dependent manner.

Different studies have demonstrated that chitosan and chitosan-essential oil coatings can disrupt filamentous fungi cell membranes due to hydrogen bonds, electrostatic and hydrophobic interactions between chitosan-essential oil components and cell membranes [56]. In addition, the barrier effect of CS + EO coatings against moisture, ethylene loss, decreasing respiration rate, and microbial attack prevents the deterioration of the organoleptic properties [57].

#### 2.2.2. Disease Damage Incidence and Infection Index

As shown in Table 2, on the 21st day of storage, the disease damage incidence was 100% in uncoated pears and pears coated with CS + RGEO 0%, while treatment with CS + RGEO 0.5% reduced the disease incidence by about 33%. On the contrary, CS + RGEO 1.0% and 1.5% totally inhibited the disease damage incidence in pears. Significant differences (*p* < 0.05) were observed between F4 and F5 concerning F1, indicating a protective effect of the CS + RGEO against fungal diseases. Regarding McKinney’s index of decay, it was significantly decreased compared to the control, by 58, 83 and 100% by F2, F3, F4, and F5, respectively. The incidence and severity of pear fruit damage decreased with increased RGEO concentrations. In previous studies, treatments with chitosan when mixed with RGEO also reduced damage of papaya [16], tomato [21], gooseberries [20], and guava fruit [19] when they were applied at the post-harvest stage. RGEO showed an effect on spore germination, increasing the lag phase and damaging the membrane of *Colletotrichum gloeosporioides*, which could explain the protective effect of the oil [16].

#### 2.2.3. Microbiological Analysis

As observed in Figure 9a, CS coatings show efficacy to inhibit the growth of the mesophilic bacteria, yeast, and molds present on pears’ surfaces. During day 12 of storage treatment, coatings F2 (CS) and F3 (CS + RGEO 0.5%) inhibited the bacteria growth by about 0.8 and 2.7 Log CFU/g after treatment. Coatings with 1.0% and 1.5% of RGEO (F4 and F5) showed more potent inhibition of about 2.8 and 3 Log CFU/g, respectively; no significant differences were evidenced between CS + RGEO formulations, demonstrating that there is no dependence on the essential oil concentration. As shown in Figure 9, there were no significant differences (*p* < 0.05) between the 6th and 12th day for F3 and F4, and they also presented similar inhibition to F5, which is advantageous since a lower amount of RGEO could be used in the formulation with an excellent mesophilic bacteria inhibition during more storage time. In addition, F3 and F4 presented no significant differences in respect to factor time from day 6, suggesting that one of these two formulations could be the most suitable from the microbiological point of view. Similar results were reported in tomatoes [21] and guava [19] coated with CS + RGEO. Usually, the mesophilic bacteria count describes the population of coccus, bacillus, and spiral bacteria present in the fruit; when the levels are high, this could indicate poor hygienic conditions and the decay of fruits [21]. Many studies have reported the antimicrobial effect of CS against Gram-positive and Gram-negative bacteria [18], which can be reinforced with the antimicrobial activity present in essential oils; the efficacy of RGEO has been reported against bacteria such as *Escherichia coli*, *Bacillus cereus*, *Pseudomonas aeruginosa,* and *Staphylococcus aureus* [16].

The statistical analysis indicated a significant difference (*p* < 0.05) during the storage time for F1 (Figure 9 B). However, F2 was statistically equal between days 3 and 6, and 9 and 12. From day 3, the formulations presented significant differences concerning the control; on day 9, F4 had significant differences regarding F2, showing the RGEO inclusion effect. On the last day of storage time, pears coated with F2 (CS) only presented a reduction of 1.2 Log CFU/g on day 12, F3 (CS + RGEO 0.5%), and F4 (CS + RGEO 1.0%) demonstrated 2.4 and 2.55 Log CFU/g reductions, respectively. F5 (CS + RGEO 1.5%) presented a reduction of 3.3 Log CFU/g in mold growth, indicating a potent antifungal effect dependent on the RGEO amount in the coating. Previous studies with CS + RGEO coatings in tomatoes reported a reduction of 2 Log CFU/g in mold growth after 12 days of storage with RGEO 0.5% [21]. Xu et al. [58] reported the better inhibition of mold growth in pear with chitosan-coatings reinforced with cinnamon oil compared to that obtained with just chitosan. As stated above, *Penicillium expansum*, *Botrytis cinerea*, and *Mucor piriformis* are the primary fungi affecting pears in the pre- and post-harvest stages, contributing to the infection and physical damage produced in the pears [4]. It has been previously demonstrated that CS has a potent antifungal effect, which helps control fungi infections [18]. It has been previously shown that chitosan coatings incorporated with various essential oils inhibited or controlled the growth of fungal agents and allowed the extent of fruits’ stability time of storage [59]. In general, a decrease in the inhibitory effect of CS + RGEO when the storage time increased was observed. In this context, de Oliveira et al. 2018 [60] suggested that during fruit ripening, the susceptibility to pathogenic fungi increases. Additionally, it could be possible that RGEO volatilization contributes to decreasing the efficacy of the film-forming dispersions.

### 2.3. Sensory Properties

It is essential to analyze the changes in color, aroma, flavor, firmness, and chemical composition during maturation due to their pivotal role in the synthesis of carotenoids and chloroplasts [61]. Sensory analysis of the coated samples is fundamental as it determines the acceptability of the final consumer of fruits. Figure 10 shows the pears’ sensory evaluation with hedonistic results during day 0, day 5, and day 10, respectively.

After treatment, sensory assessors presented significant differences between control and coated pears with F4 (CS + RGEO1.0%) and F5 (CS + RGEO1.5%) after day 5. On day 5, untreated samples were not accepted by most consumers, indicating significant acceptability of the pears treated with CS + RGEO 1.0% emulsion and CS + RGEO 1.5%. Flavor, aroma, and texture were the attributes that had a significant influence on the assessor’s tests, while gloss and color were less accepted for the pears with F4 and F5, without significant differences between F3 and F4, respectively.

Palatability is one of the most critical organoleptic properties of foods, and it is mainly dependent on the balance between organic acids (acidity) and sugar (sweetness), indicating the importance of the metabolic pathway of the fruit for its value [30]. One of the biggest problems with using essential oils in edible coatings is the transfer of compounds from the oil to the fruit, which can change the product’s palatability [16]. As previously evidenced, the CS-RGEO coatings delayed the pears’ ripening; this could influence the coated samples’ acceptance compared to the uncoated ones. In our study, CS-RGEO coatings did not show flavor transfers according to the sensory assessors’ opinions. Similar results were obtained in papaya [16], guava [19], and fresh-cut pears [22], where fruit treated with chitosan-based coatings reinforced with essentials oils or natural extracts received better acceptance than uncoated fruit.

## 3. Materials and Methods

### 3.1. Fruit Samples

Three hundred and sixty healthy Packham’s Triumph pears (*Pyrus communis* L.) with visual uniformity in color and size and no physical, microbiological or mechanical damage were selected in a maturation stage of four according to the Colombian technical standard (NTC) 1291 from a local market of Jamundí, Valle del Cauca, Colombia (3°15′39″ N 76°32′22″ O), and they were stored at a cold temperature (18 °C) before use.

### 3.2. Preparation of Edible Coatings

The preparation of the film-forming emulsions (FFE) based on CS + RGEO followed the reported methodology [19], mixing chitosan (degree of deacetylation = 85%, Mw = 190.000–310.000 Da, Sigma-Aldrich, Palo Alto, California, USA) in aqueous acid solution (1% acetic acid) to obtain a 2% (*w* of CS/*v*) solution. Then, 2.5% (*v*/*v*) of glycerol was used as a plasticizer and Tween 80 (1% *v*/*volume* RGEO) as a surfactant followed to complete homogenization. RGEO was added to reach final concentrations of 0.5%, 1.0%, and 1.5% (*v*/*volume* emulsion). Finally, the emulsions were degassed.

### 3.3. Coating of the Fruits

Pears were washed with 0.5% (*v*/*v*) of CECURE^®^, a cetyl pyridinium chloride solution (Safe Foods, Rogers, AR, USA), and sterile distilled water. Then, they were left to dry at 18 °C. Four batches of 90 pears were successively coated with the formulations, which consisted of one control (consisting of a pear dipped in pure distilled water, F1) and four different formulations (F2 = CS, F3 = CS + RGEO 0.5%, F4 = CS + RGEO 1.0%, and F5 = CS + RGEO 1.5%).

Freshly prepared CS + RGEO emulsions were applied on the pear surfaces using a simple prototype designed to process three pears simultaneously through the dip-coating technique, allowing a uniform coating and less contact with the fruit before the drying stage. The prototype consisted of a movable part secured by a clamp holder to a modified retort stand and 500 mL beakers (4.9 in × 3.9 Ø) holding pears off their stalk with a mini spring clamp shown in Figure 11. Successively, they were dried for one hour at a temperature of 18 °C ± 2 °C. The coated samples were placed in Polyethylene Terephthalate (PET) boxes and stored at a temperature of 18 °C ± 2 °C and relative humidity of 70%, in a cabinet with a protective mesh.

Fifteen pears of each treatment were selected periodically for the analysis. Evaluations of pears’ physical-chemical and mechanical properties occurred at days 0, 3, 7, 10, 14, and 21. Microbiological analysis occurred during days 0, 3, 6, 9 and 12. Sensorial properties were measured at days 0, 5 and 10.

### 3.4. Post-Harvest Quality Properties of Pears

In order to evaluate post-harvest quality properties of pears, several physical-chemical parameters were performed according to Grande Tovar et al. [19]:

#### 3.4.1. pH and Soluble Solids (SS)

A calibrated potentiometer (Orion Star ^TM^ A11, Thermo Fisher Scientific, Waltham, MA, USA) was used for the pH measurements. Ten grams of fruit were homogenized in 100 mL of distilled water for this purpose. Soluble solid contents (SS) (%) were determined using a portable refractometer (Brix 0–90% Model 3090, BRIXCO).

#### 3.4.2. Titratable Acidity (TA)

Five grams of fruit in 50 mL of distilled water were homogenized for each treatment. The acidity (citric acid %) was determined using potentiometric titration with 0.1 N NaOH, according to Equation (1):(1)$$Citric\;acid\;\left(\%\right)=\left(\frac{V_{1}-N}{V_{2}}\right)\times K\times 100$$
where V_1_ is the volume of NaOH used (mL), V_2_ is the volume of the sample (mL), k is the equivalent weight of citric acid (0.064 g/meq), and *N* is the normality of NaOH (0.1 meq/mL).

#### 3.4.3. Maturity Index (MI)

Maturity index (MI) was calculated using Equation (2):(2)$$MI=\frac{\%BRIX}{\%ACID}.$$

#### 3.4.4. Weight Loss Percentage

Fruits were weighed using a scale (Aviator 7000, OHAUS, Parsippany, NJ, USA) on day 0, 3, 7, 10, 14, and 21 of storage. The weight loss was evaluated as the difference between initial and final fruit weight that storage interval expressed as a weight loss percentage on a fresh weight basis.

#### 3.4.5. Color Analysis

The analysis of the fruit surfaces’ color was performed with a colorimeter (CM-600d, Konica Minolta Optics Inc., Tokyo, Japan) and with the CIELab color coordinates (L*, a*, and b*), following the parameters proposed by Grande Tovar et al. The reported values correspond to the average of three measurements in each treatment in the equatorial zone of the fruits at three random points.

#### 3.4.6. CO_2_ Respiration Rate

The equipment employed was an EcoChamber ME-6667 (PASCO, Roseville, CA, USA) including a carbon dioxide sensor PS-2110 used to measure the CO_2_ levels using a gaseous CO_2_ analyzer during days 0, 3, 7, 10, 14 and 21 of the coating process.

#### 3.4.7. Mechanical Properties of Pears

Compression resistance (kPa), Force (N), and Deformation (%) were evaluated after the penetration of fruits using a cylindrical penetrometer (3 mm diameter) coupled with an EZ-Test (EZ-LX, Shimadzu-USA) texturometer. A penetration speed of 5 mm/s was used. The average penetration on the equatorial zone of the fruits at three random points was reported.

### 3.5. Decay Evaluation

#### 3.5.1. Antimicrobial Assay

The microorganism count was performed by two replicates at days 0, 3, 6, 9, and 12 using a reported methodology [35]. Total viable count (CFU/mL) of aerobic mesophylls was realized using a Plate Count Agar (Difco, Kansas City, MO, USA) after incubation at 25 °C for 48 h. For the mold and yeast count, potato-glucose agar (PDA) was used as media incubated at a temperature of 25 ± 0.2 °C for five days.

#### 3.5.2. Severity Index (SI)

Severity index (SI) was evaluated visually according to the methodology proposed previously [19], following the scale of Figure 12. The results of physical or microbial damage were determined with Equation (3):(3)$$Severity\;Index=\frac{1n+2n+3n+4n}{N}$$
where *n* = number of fruits classified in each level of the damage scale and *N* = number of total fruits analyzed in each treatment per day. The decay index of fruits was evaluated on days 0, 3, 7, 10, 14, and 21.

#### 3.5.3. Disease Damage Incidence

The incidence of fungal diseases was measured following the method reported before [21]. The disease damage incidence was calculated using Equation (4):$$Disease\;damage\;incidence\;\left(\%\right)=\frac{number\;of\;infected\;pears}{total\;samples\;per\;treatment}\times 100$$

#### 3.5.4. Infection Index

The infection index is also known as the McKinney Index, which correlated the disease damage incidence with the severity and was determined following the method reported by [62] using Equation (5):(4)$$McKinney\;Index\;\left(\%\right)=\left[\sum \frac{\left(d\times f\right)}{\left(n\times D\right)}\right]\times 100$$

Here, d is the category of severity on the fruit, f is its frequency, n is the total number of examined fruits, and D is the highest category of severity index that occurred on the hedonic scale.

### 3.6. Sensorial Activity

The test was carried out with one group of 50 non-trained assessors made up of women and men in equal numbers with an age range between 19 and 26 years during days 0, 5, and 10 of storage [21]. All the tests were performed in the morning hours. The samples were cut into cubes of the same size. Then, the peel was not removed by the pear consumers’ preferences in Colombia. The assessors were informed of the methodology and signed an informed consent with the test details. Attributes such as color, flavor, aroma, texture, and gloss were evaluated in the test. Assessors were asked to score the difference between samples, where 0–2 represented extreme dislike; 3–5 fair; 6–8 good; and 9 excellent for each attribute.

### 3.7. Statistical Analysis

A completely randomized design was applied in this study. All experiments were conducted in triplicates except for CO_2_ respiration rate (*n* = 2). The data from each sampling point are shown as the mean ± SD and were statistically evaluated by ANOVA, followed by individual comparisons using Duncan’s Multiple Range Test, with a confidence level of 95% (α = 0.05). This test was used to assess the effect of diverse formulations on the response variables described previously for each time point. The Fisher’s Least significant differences (LSD) test was used to assess the differences between days for each treatment.

## 4. Conclusions

In the present work, we demonstrated a protective effect on the quality properties of pears after the surface application of chitosan-*Ruta graveolens* (CS + RGEO) essential oil coatings in four different concentrations (0%, 0.5%, 1.0%, and 1.5%, *v*/*v*) during 21 days of storage under 18 °C. The fruits’ physical-chemical characteristics were analyzed, evidencing a protecting effect of coatings against the ripening process; despite no significant differences between CS and CS-RGEO formulations, there is no affectation on the properties evaluated. Maturity index, decay index, disease damage incidence, and color results correspond to less ripe fruits. A weight loss reduction of 50% (from 40.2 ± 5.3 to 20.3 ± 3.9) compared to the uncoated pears was evident with CS + RGEO 0.5%, demonstrating a barrier effect of the coatings. After mechanical property analysis for the coated and uncoated fruits, the pears’ highest mechanical resistance was obtained with CS + RGEO 0.5% after 21 days of storage, both for compression resistance (7.42 kPa) and force (22.7 N).

The McKinney index indicated total protection for pears coated with CS + RGEO 1.5%. Formulations including 15 μL/mL of RGEO significantly reduced aerobic mesophilic bacteria by 3.0 Log CFU/g and molds and yeast by 3.3 Log CFU/g compared to the control, without affecting consumer perception.

The sensorial analysis of the coated fruits demonstrated that all the formulations were acceptable for the organoleptic attributes and were suitable for human consumption. The study showed that the formulations including CS + RGEO 0.5% are suitable for post-harvest treatment for pears and show adequate antimicrobial protection with a lower oil concentration (which would be convenient in economic terms), improving consumers’ acceptance.

## Figures and Tables

**Figure 1 molecules-26-00725-f001:**
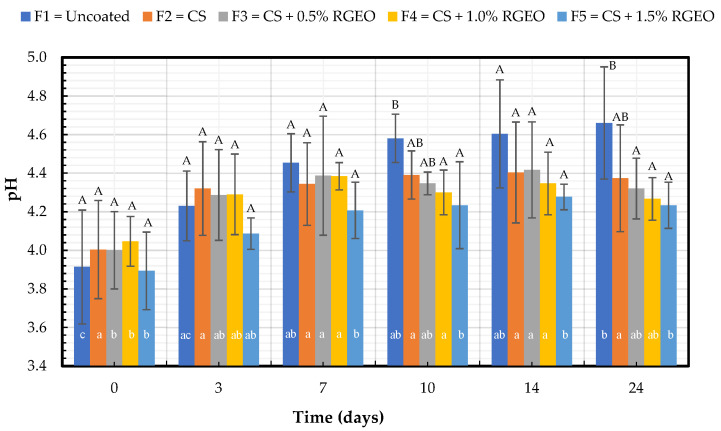
Evolution of pH in pears with CS + RGEO treatments: F1 = uncoated, F2 = CS, F3 = CS + RGEO 0.5%, F4 = CS + RGEO 1.0%, and F5 = CS + RGEO 1.5%. According to the ANOVA test, mean values and intervals of Duncan test for treatments and Fisher for days with 95%. Different lowercase letters (a, b, c) indicate significant differences between days according to the Least Significant Differences (LSD) Fisher test for each treatment in a confidence interval of 95%. In contrast, different uppercase letters (A, B, C) indicate significant differences between treatments for each day according to the Duncan test in a confidence interval of 95%.

**Figure 2 molecules-26-00725-f002:**
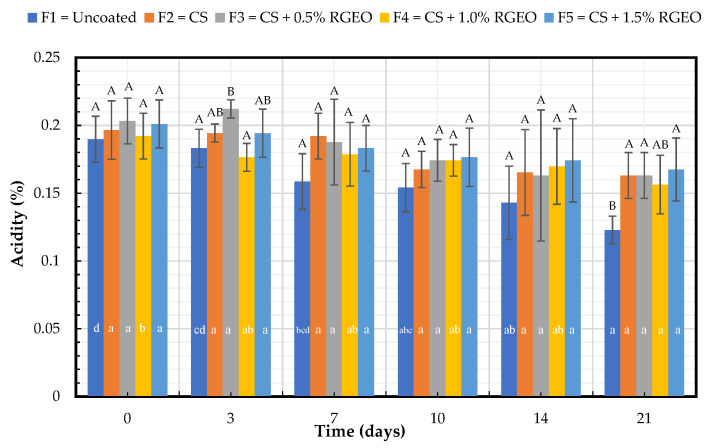
Evolution of the percentage of titratable acidity expressed as citric acid in pears with CS + RGEO treatments: F1= uncoated, F2 = CS, F3 = CS + RGEO 0.5%, F4 = CS + RGEO 1.0%, and F5 = CS + RGEO 1.5%. According to the ANOVA test, mean values and intervals of Duncan test for treatments and Fisher for days with 95%. Different lowercase letters (a, b, c) indicate significant differences between days according to the LSD Fisher test for each treatment in a confidence interval of 95%. In contrast, different uppercase letters (A, B, C) indicate significant differences between treatments for each day according to the Duncan test in a confidence interval of 95%.

**Figure 3 molecules-26-00725-f003:**
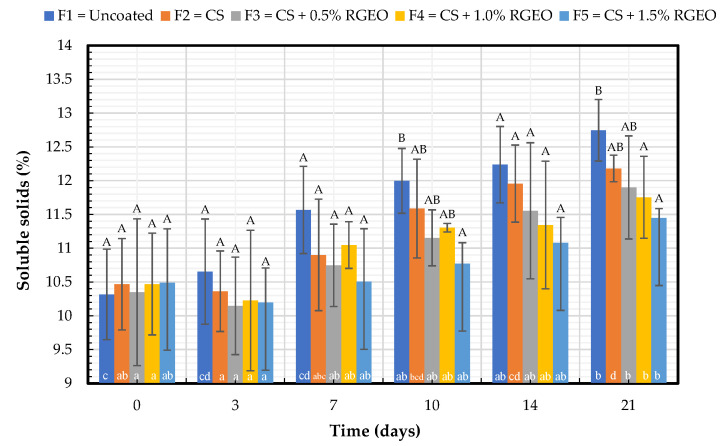
Evolution of the total soluble solids content by total de SS measurement pears with CS + RGEO treatments: F1 = uncoated, F2 = CS, F3 = CS + RGEO 0.5%, F4 = CS + RGEO 1.0%, and F5 = CS + RGEO 1.5%. According to the ANOVA test, mean values and intervals of Duncan test for treatments and Fisher for days with 95%. Different lowercase letters (a, b, c) indicate significant differences between days according to the LSD Fisher test for each treatment in a confidence interval of 95%. In contrast, different uppercase letters (A, B, C) indicate significant differences between treatments for each day according to the Duncan test in a confidence interval of 95%.

**Figure 4 molecules-26-00725-f004:**
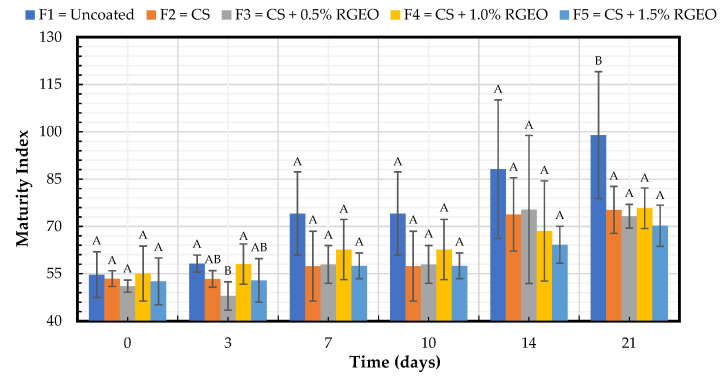
Maturity index of pears during the storage time with CS + RGEO treatments: F1 = uncoated, F2 = CS, F3 = CS + RGEO 0.5%, F4 = CS + RGEO 1.0%, and F5 = CS + RGEO 1.5%. According to the ANOVA test, mean values and intervals of Duncan test for treatments and Fisher for days with 95%. Different lowercase letters (a, b, c) indicate significant differences between days according to the LSD Fisher test for each treatment in a confidence interval of 95%. In contrast, different uppercase letters (A, B, C) indicate significant differences between treatments for each day according to the Duncan test in a confidence interval of 95%.

**Figure 5 molecules-26-00725-f005:**
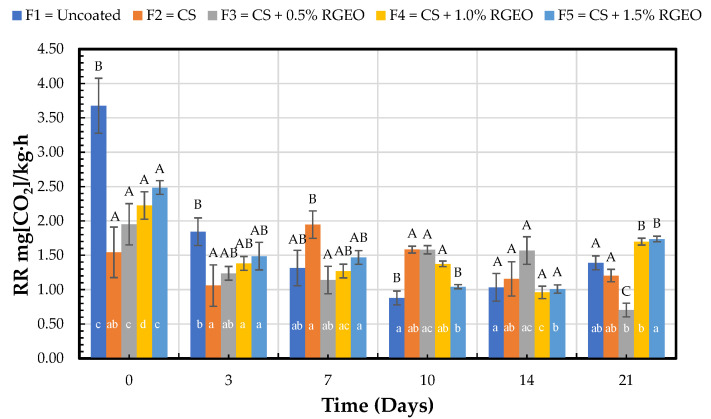
CO_2_ respiration rate in pears with CS + RGEO treatments: F1 = control, F2 = CS, F3 = CS + RGEO 0.5%, F4 = CS + RGEO 1.0%, and F5 = CS + RGEO 1.5%. According to the ANOVA test, mean values and intervals of Duncan test for treatments and Fisher for days with 95%. Different lowercase letters (a, b, c) indicate significant differences between days according to the LSD Fisher test for each treatment in a confidence interval of 95%. In contrast, different uppercase letters (A, B, C) indicate significant differences between treatments for each day according to the Duncan test in a confidence interval of 95%.

**Figure 6 molecules-26-00725-f006:**
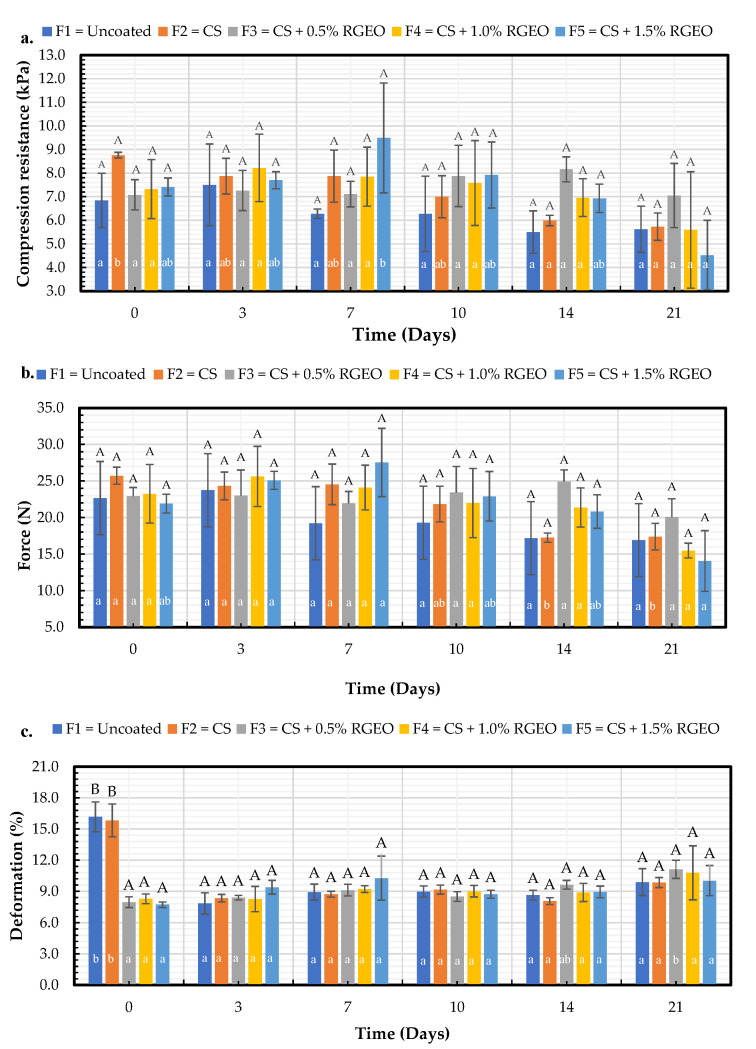
Evolution of: (**a)** Compression resistance (kPa), (**b**) Force (N) and (**c**) Deformation (%), in pears with chitosan (CS) and oil treatments (RGEO): F1 = control, F2 = CS, F3 = CS + RGEO 0.5%, F4 = CS + RGEO 1.0%, and F5 = CS + RGEO 1.5%. According to the ANOVA test, mean values and intervals of Duncan test for treatments and Fisher for days with 95%. Different lowercase letters (a, b, c) indicate significant differences between days according to the LSD Fisher test for each treatment in a confidence interval of 95%. In contrast, different uppercase letters (A, B, C) indicate significant differences between treatments for each day according to the Duncan test in a confidence interval of 95%.

**Figure 7 molecules-26-00725-f007:**
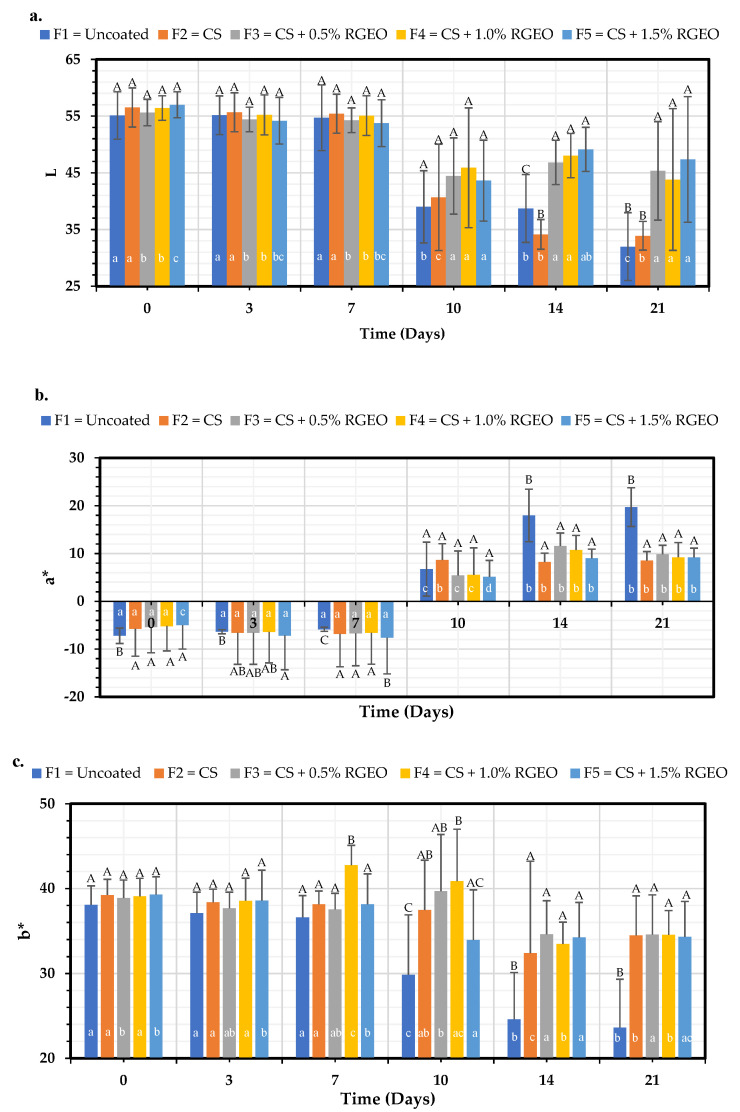
Color evolution where (**a**) L represented the lightness of the color, (**b**) a* its position between red and green and (**c**) b* its position between yellow and blue during 24 days of pears with CS + RGEO treatments: F1 = control, F2 = CS, F3 = CS + RGEO 0.5%, F4 = CS + RGEO 1.0%, and F5 = CS + RGEO 1.5%. According to the ANOVA test, mean values and intervals of Duncan test for treatments and Fisher for days with 95%. Different lowercase letters (a, b, c) indicate significant differences between days according to the LSD Fisher test for each treatment in a confidence interval of 95%. In contrast, different uppercase letters (A, B, C) indicate significant differences between treatments for each day according to the Duncan test in a confidence interval of 95%.

**Figure 8 molecules-26-00725-f008:**
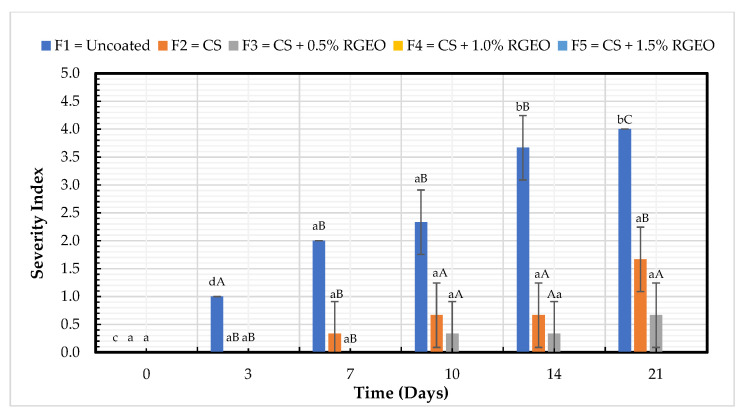
Evolution of the decay index in pears with chitosan (CS) and treatments of oil (RGEO): F1 = control, F2 = CS, F3 = CS + RGEO 0.5%, F4 = CS + RGEO 1.0%, and F5 = CS + RGEO 1.5%. According to the ANOVA test, mean values and intervals of Duncan test for treatments and Fisher for days with 95%. Different lowercase letters (a, b, c) indicate significant differences between days according to the LSD Fisher test for each treatment in a confidence interval of 95%. In contrast, different uppercase letters (A, B, C) indicate significant differences between treatments for each day according to the Duncan test in a confidence interval of 95%.

**Figure 9 molecules-26-00725-f009:**
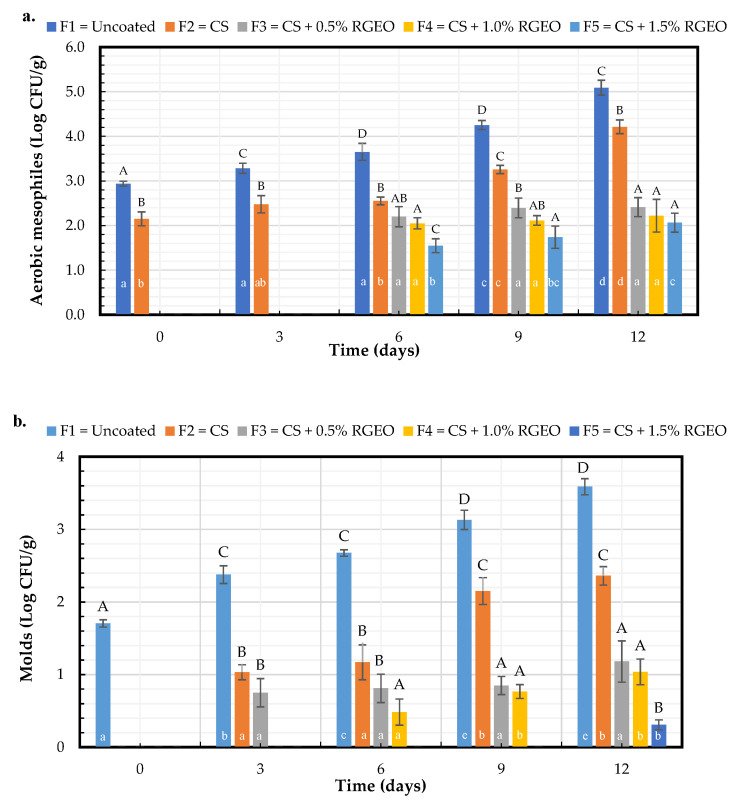
Effect of treatments on the concentration of: (**a**) aerobic mesophilic, (**b**) counting of molds and yeasts in pears with CS + RGEO treatments: F1 = uncoated, F2 = CS, F3 = CS + RGEO 0.5%, F4 = CS + RGEO 1.0%, and F5 = CS + RGEO 1.5%. According to the ANOVA test, mean values and intervals of Duncan test for treatments and Fisher for days with 95%. Different lowercase letters (a, b, c) indicate significant differences between days according to the LSD Fisher test for each treatment in a confidence interval of 95%. In contrast, different uppercase letters (A, B, C) indicate significant differences between treatments for each day according to the Duncan test in a confidence interval of 95%.

**Figure 10 molecules-26-00725-f010:**
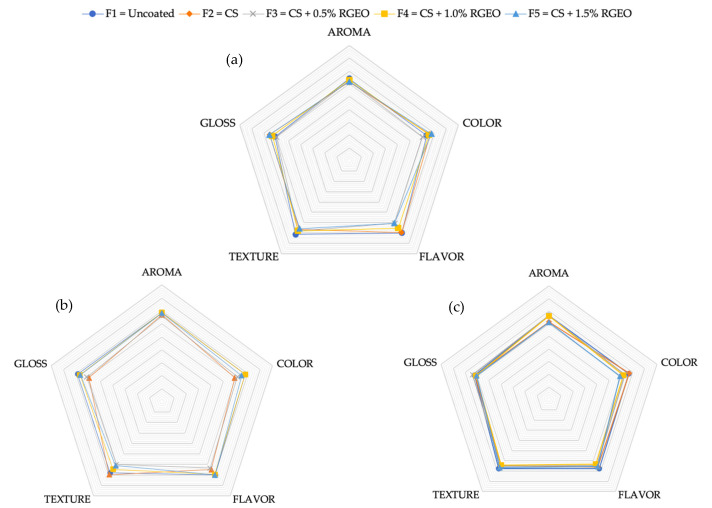
The hedonistic scale of guavas treated with the different formulations on day 0 (**a**), day 5 (**b**), and day 10 (**c**).

**Figure 11 molecules-26-00725-f011:**
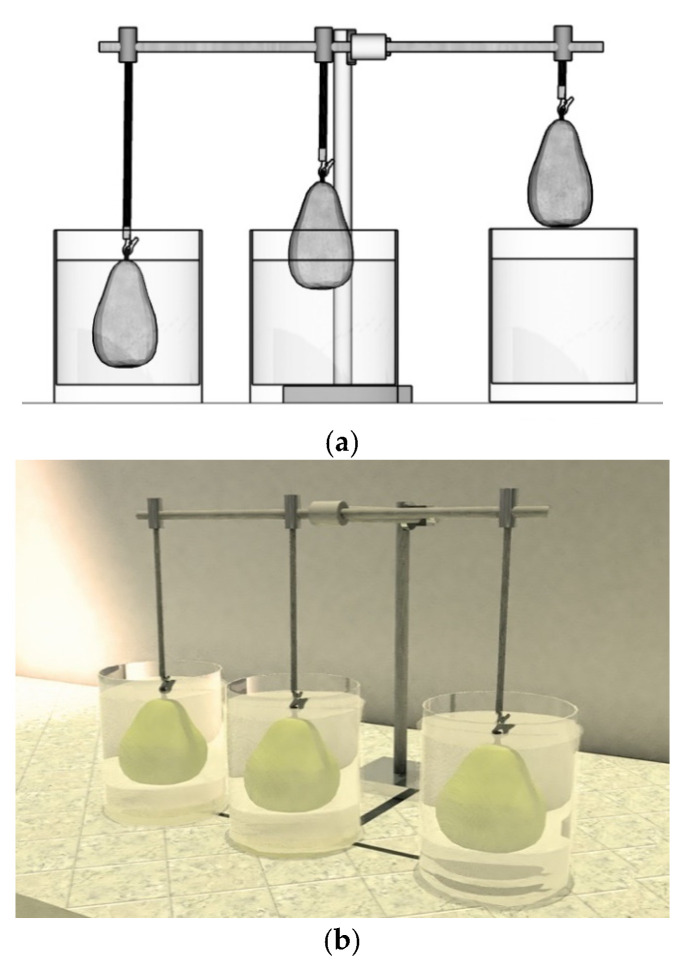
Dip-coating prototype. (**a**) A schematic view of pears’ dip-coating process, (**b**) a detailed picture of the prototype.

**Figure 12 molecules-26-00725-f012:**
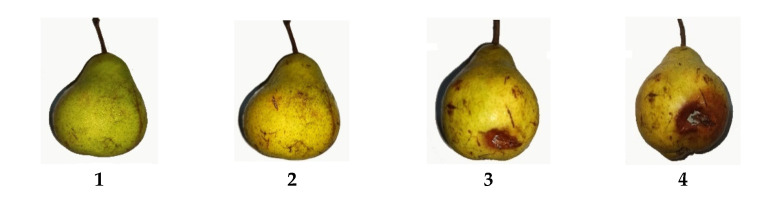
Empirical damage scale used for scale of the pear, where 1 = no damage (0% damage), 2 = mild damage (10–15% damage), 3 = moderate damage (25–50% damage), and 4 = severe damage (>50% damage).

**Table 1 molecules-26-00725-t001:** Evolution of weight loss percentage in pears with CS + RGEO treatments: F1 = control, F2 = CS, F3 = CS + RGEO 0.5%, F4 = CS + RGEO 1.0%, and F5 = CS + RGEO 1.5%.

Days	0	3	7	10	14	21
Treatment	Weight Loss (%)					
Control	0	5.4 ± 1.8 ^aA^	10.4 ± 1.0 ^abA^	15.6 ± 2.6 ^bA^	23.2 ± 1.7 ^cA^	40.2 ± 5.3 ^dB^
CS + 0% RGEO	0	4.4 ± 0.4 ^aA^	9.7 ± 0.8 ^abA^	14.7 ± 1.3 ^bcA^	21.5 ± 0.8 ^cAB^	29.8 ± 9.6 ^dAB^
CS + 0.5% RGEO	0	5.7 ± 1.3 ^bA^	9.4 ± 1.4 ^cA^	15.2 ± 0.8 ^dA^	21.9 ± 0.8 ^aA^	20.3 ± 3.9 ^aA^
CS + 1.0% RGEO	0	4.4 ± 0.2 ^bA^	9.2 ± 0.5 ^bcA^	15.5 ± 2.3 ^acA^	21.6 ± 1.6 ^aAB^	21.3 ± 8.5 ^aA^
CS + 1.5% RGEO	0	4.4 ± 1.1 ^aA^	9.3 ± 0.8 ^abA^	14.7 ± 1.0 ^bcA^	16.6 ± 5.3 ^cB^	23.8 ± 5.6 ^dA^

According to the ANOVA test, mean values and intervals of Duncan test for treatments and Fisher for days with 95%. Different lowercase letters (a, b, c) indicate significant differences between days according to the LSD Fisher test for each treatment in a confidence interval of 95%. In contrast, different uppercase letters (A, B, C) indicate significant differences between treatments for each day according to the Duncan test in a confidence interval of 95%.

**Table 2 molecules-26-00725-t002:** Damage incidence and McKinney index of weight in pears with CS + RGEO to the final day of storage time.

Treatment	Incidence (%)	McKinney Index
F1: Uncoated	100 ± 0.0 ^A^	100 ± 0.0 ^C^
F2: CS + 0% RGEO	100 ± 0.0 ^A^	41.7 ± 14.4 ^B^
F3: CS + 0.5% RGEO	66.7 ± 40 ^A^	16.7 ± 14.4 ^C^
F4: CS + 1.0% RGEO	0.0 ± 0.0 ^B^	0.0 ± 0.0 ^C^
F5: CS + 1.5% RGEO	0.0 ± 0.0 ^B^	0.0 ± 0.0 ^C^

According to the ANOVA test, mean values and intervals of the Duncan test for treatments. According to the Duncan test, different uppercase letters (A, B, C) indicate significant differences between treatments for each day in a confidence interval of 95%.

## Data Availability

The data presented in this study are available in the article or Appendix A.

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
