# Peer review of "Packham’s Triumph Pears (Pyrus communis L.) Post-Harvest Treatment during Cold Storage Based on Chitosan and Rue Essential Oil"

_molecules, 2021, doi:10.3390/molecules26030725_

Round 1

Reviewer 1 Report

This experiment was done to determine if chitosan and essential oils will help prolong quality of an heirloom pear.  

References in this paper are excessive.  50 is plenty given the scope of the work.

Overall, there are some interesting trends showing success with chitosan and RGEO  but the graphs are extremely hard to follow as presented.  There is no indication of the experimental design used; surely the first analysis of data should be relative differences of treatments within a time point.  For instance, in Figure 6, where maturity index (which is merely the ratio of ssc/titr. acid), Rgeo 0.5, 1.0 and 1.5 do not appear to be different from each other but are presented instead as differences by day within a treatment.  This to me gives misleading information about the relative effectiveness of REGEO concentrations.  Unfortunately, Dunnets and duncans analysis are not presented in the statistic section but instead as a figure heading.

Other areas for improvement: introduction should really be about this particular pear cv first, then chitosan then essential oils.

Chitosan slows respiration primarily through biological effects or by coating stomates and slowing respiration physically (line 256). Fig 8-is there a statistical difference in respiration?

line 480: isn't yellow color from chlorophyll degradation first? can't read the reference.

Overall, this paper needs a massive rewrite to focus on the more meaningful results and cut the verbiage down.  Statistical analysis may not be appropriate for this design. 

Author Response

Reviewer 1

This experiment was done to determine if chitosan and essential oils will help prolong quality of an heirloom pear.  

References in this paper are excessive.  50 is plenty given the scope of the work.:

R// We appreciate the reviewer's comment. The references were reduced according to the adjustment realized in the manuscript.

Overall, there are some interesting trends showing success with chitosan and RGEO  but the graphs are extremely hard to follow as presented.  There is no indication of the experimental design used; surely the first analysis of data should be relative differences of treatments within a time point.  For instance, in Figure 6, where maturity index (which is merely the ratio of ssc/titr. acid), Rgeo 0.5, 1.0 and 1.5 do not appear to be different from each other but are presented instead as differences by day within a treatment.  This to me gives misleading information about the relative effectiveness of REGEO concentrations.  Unfortunately, Dunnets and duncans analysis are not presented in the statistic section but instead as a figure heading.

R// We appreciate the reviewer's comment. The graphs were adjusted to facility their interpretation. The experimental design type used was included (lines 194-199):

A completely randomized design was applied in this study. All experiments were conducted in triplicates except for CO2 respiration rate (n = 2). The data from each sampling point are shown as the mean ± SD and were statistically evaluated by ANOVA, followed by individual comparisons using Duncan's Multiple Range Test, with a confidence level of 95% (α = 0.05). This test was used to assess the effect of diverse formulations on the response variables described previously for each time point. The Fisher test was used to assess the differences between days for each treatment.

The Duncan test changed the Dunnet test to validate the differences between treatments for each time point. This change allowed to deepen the effect of RGEO in coatings.

Other areas for improvement: introduction should really be about this particular pear cv first, then chitosan then essential oils.

R// We appreciate the reviewer's comment. The introduction was adjusted following the suggestions (lines 38-80).

Pears (Pyrus communis L.) is one fruit with high production figures in Latin America, with around one million tons annually [1]. China is the major producer in Asia with about 56 million tons, while Italy does the same in Europe, followed by the United States, Spain, and Turkey [2]. The immense worldwide production of pears also brings significant losses, with farmers and producers as the main affected supply chain sectors. Pear shelf-life depends on several factors such as growing season climate, nutrition, environmental conditions, cultivar, and microbial attacks, causing losses in the pear industry can be range from 5% and 50% of the production [2,3]. The primary post-harvest diseases in pear, such as blue mold, pink mold, grey mold, and black spot, are related mainly to fungi of the genus Mucor piriformis, Penicillium expansum, and Botrytis cinerea [4].

Pear fruits are climacteric, and the coating is considered one of the most popular techniques to prolong shelf life [5]. Several studies have evaluated the effect of edible coatings containing polysaccharides, such as soy protein isolate (SPI), in combination with hydroxypropyl methylcellulose (HPMC) and olive oil on 'Babughosha' Pears (Pyrus communis L.) stored at ambient temperature (28 ± 5 °C and 60 ± 10% RH). Various edible coatings like zein-oleic acid [6], plant oils [7], Shellac, Semperfresh, carboxymethylcellulose [8] are useful on pears.

Chitosan, a deacetylated derivative of chitin, is deeply investigated for many industrial and medical applications [9]. Chitosan coatings demonstrated an improvement in the storability of several perishable fruits. Barrier properties against moisture and oxygen and a capacity to preserve volatile bioactive compounds as well as a potent antimicrobial and food quality preservation capacity have been observed [10]. Some authors have evaluated chitosan coatings on pears obtaining a significantly delayed ripening and improved fruit firmness [11,12]. Nonetheless, chitosan coatings present a disadvantage its hydrophilic nature forces the introduction of hydrophobic composites such as some essential oils, which also provide antioxidant and antimicrobial properties to the food during the post-harvest stage [13].

On the other hand, the treatment of crops with essential oils (EO) could be a safer and alternative option for post-harvest preservation. Essential oils (EO) are complex mixtures of terpenes and terpenoids, coumarins, and homologs of phenylpropanoids, with biological properties and sometimes strong odors. Ruta graveolens plant has been identified as a source of different antifungal compounds such as 5- and 8-methoxy psoralen, with potent activity against fungi like  Rhizoctonia solanii, Fusarium spp., Pyrenochaeta lycopersici, Trichoderma viride, and Penicillium spp. [14]. In previous studies, the RGEO has shown high activity against the fungi Aspergillus fumigatus, Cladosporium herbarum, Candida albicans, and Colletotrichum gloeosporioides [15,16]. However, despite its antimicrobial activity, EOs are volatile and could influence the food's organoleptic properties with the possible presence of phytotoxicity [17]. A more recent and useful strategy uses composites of biopolymers and essential oils as coatings on the fruit surfaces for a controlled release of the volatile compounds inhibiting microbial growth on the fruit surface and conserving for more time nutritional composition, quality, and product acceptance [18].

Despite all the information above, no literature using CS+RGEO coatings to preserve pears' post-harvest decay is available. Taking into account the antifungal activity of RGEO components and the efficiency of CS+RGEO maintaining guava [19], cape gooseberries [20], papaya [16], and tomato [21] quality properties while inhibiting microbial growth, the objective of this study was to evaluate the effect of the application of CS+RGEO coatings on Packham's Triumph pears to preserve their quality during cold storage.

Chitosan slows respiration primarily through biological effects or by coating stomates and slowing respiration physically (line 256). Fig 8-is there a statistical difference in respiration?

R// We appreciate the reviewer's comment. The statistical differences in the graph were included both in the graph and in the discussion of the results (lines 363-369):

Figure 7 shows the high respiration rate of uncoated fruits at the beginning of the study compared to coated fruits presenting significant differences. After three and seven days, there was an apparent decrease in the respiration rate due to the treatments. On day 14, no significant differences were observed between treatments. On the last day, F3 has a significant decrease concerning the control and the other formulations. Besides, it showed significant differences regarding factor time in day 21 compared to day 0. F4 and F5 had a higher respiration rate of the treatments.

line 480: isn't yellow color from chlorophyll degradation first? can't read the reference.

R// We appreciate the reviewer's comment. We added that the yellow color is also a consequence of chlorophyll degradation. The reference was adjusted in the new version, and it should be easily read.

Overall, this paper needs a massive rewrite to focus on the more meaningful results and cut the verbiage down.  Statistical analysis may not be appropriate for this design. 

R// We appreciate the reviewer's comment. The manuscript was adjusted, starting with a change in one of the statistical analyses, with the respective modification of the study's results and the discussion. Besides, a simplification was made in many manuscript sections.

Reviewer 2 Report

Dear Author(s)

After an exhaustive revision, the manuscript is RECONSIDER AFTER MAJOR REVISION (CONTROL MISSING IN SOME EXPERIMENTS). In general, the study is closely connected to the journal's objectives. The study is very interesting. However, the manuscript has a lot parts that need to be corrected, since results and the discussion present problems, since this section is very incomplete. Also, there are problems with the statistical results, but if the authors fix the comments, the manuscript could be accepted.

In the following pages, I give a detailed revision of the manuscript.

Best regards

GENERAL COMMENTS

Affiliations have certain errors:

- Where are the email and the initials of the authors?

In all the manuscript, all ordinal numbers should be deleted and left only as a normal number.

  1. INTRODUCTION

General comments

The introduction is good, since it starts from general to particular. The English is good. The objective of the study should more clearly. However, it is possible to denote some little problems in the order of some lines.  My observations:

- Line 40. What is the percentage of loss approximately?

- Line 43. In terms of tons, how much are "four thousand boxes"?

- Line 43. It is not necessary to place the reference [3] twice.

- The authors should restructure the introduction, since Lines 41-45 should be the beginning of the introduction, and then, start with "Several food losses ..."

- Lines 46-49 are similar to Lines 38-40. The authors must rearrange these lines.

- Line 54. "... cultivar, microbial attacks, ..."

The line should be changed to "... cultivar, and microbial attacks, ..."

- Lines 49-50. "Filamentous fungi 'growth in climacteric fruits, like pears, is a significant quality problem ..."

The authors should rewrite the line, since it creates confusion for the reader.

  • Lines 74-79. What is the intention with the information? These lines have no connection with the aforementioned.

Lines 99-108. What is the intention with the information? These lines have no connection with the aforementioned.

  1. MATERIALS AND METHODS

General comments

This part is complete and very clear. However, I have some observations:

  • In the subsections are not necessary to repeat the same reference.

  • Line 122. 2.2 Preparation of edible coatings.

The authors must indicate the amounts of material used, since "a specific amount" is insufficient to reproduce the experiments.

"followed our previous reported methodology [39]". The authors must rewrite these words.

"(Krauter, Colombia- previously characterized [39]),". The authors must delete these words.

  • Line 130. "2.1" Coating of the fruits.

This subsection corresponds to "2.3". Thus, all the following subsections should be fixed.

"of 90 pears each".

The authors should delete the word "each".

"physical-chemical and mechanical properties occurred at days 0, 3rd, 7th, 10th, 14th, and 21st. Microbiological analysis occurred during days 0, 3rd, 6th, 9th, and 12th. Sensorial properties were measured at days 0, 5th, and 10th. ".

The authors should remove ordinal numbers. The authors should leave only numbers.

Why did the authors perform the physical-chemical and mechanical properties, microbiological analysis and sensory properties at different days? This is worrying, since between day 6 and day 7 (for example) multiple changes can occur, and it would affect the integrity of the samples, and thus, it would change all results obtained before and after.

  • Line 155. 2.3 Post-harvest quality properties of Pears

The authors do not need to add subsections, since the authors used the same methodology [39].

The authors must write the characteristics of all equipment used (model, country, etc).

  • Line 182. 2.4. Decay evaluation

The authors should remove ordinal numbers. The authors should leave only numbers.

  • Line 209. 5. Sensory Activity

The authors should indicate more details regarding the average age, number of women, number of men, etc.

  1. RESULTS AND DISCUSSION

General comments

The results and discussion section should contain a description of the results, comparison with other studies and discussion of the results obtained with respect to other studies.

The authors should add the entire error bar in all the figures.

All subsections have only descriptions of results. Therefore, the authors should delve into comparison of results and discussions. The introduction that begins each subsection may help to the authors as a discussion.

  • Lines 223-227. All values must be inserted in the section supplementary material.

  • Lines 229-231. These lines should be deleted, since all this was explained in the section materials and methodology.

  • Line 232. 3.2.1. pH analysis

The Figures 3 shows problems about the statistics. For example, at day 21, the samples have different letters (F1 = b, F2 = a, F3 = ab, F4 = ab and F5 = b) and the authors indicate "not present significantly different". Thus, why the letters are different?. Additionally, the comparison between days has the same problem, since, for example, at day 14 and day 21 there are no significant differences, but the authors mention it with different letters.

  • Line 248. 3.2.2. Titratable Acidity (TA)

This section has a small mention of comparison of results with other study, but it is insufficient.

The Figure 4 presents the same problems (error bars and statistics) as mentioned in the Figure 3.

  • Line 268. 3.2.3. Soluble Solids Content (SS)

This section has a large introduction, which could well be used as a discussion.

The Figure 5 presents the same problems (error bars and statistics) as mentioned in the Figure 3.

  • Line 291. 3.2.4. Maturity Index

This section has a small discussion, but it is insufficient.

The Figure 6 presents the same problems (error bars and statistics) as mentioned in the Figure 3.

  • Line 317. 3.2.5. Weight Loss

This section has a small discussion, but it is insufficient.

The Table 1 presents a statistical problem. Example, in CS+0%RGEO, Day 7 presents "ab", but the result has significant different with the other days.

  • Line 331. 3.2.6. Severity Index (SI)

What is DI?

This section presents a good comparison with other studies, but a little discussion with the results of the authors.

The Figure 7 presents the same problems (error bars and statistics) as mentioned in the Figure 3.

  • Line 358. 3.2.7. Disease damage incidence and Infection index

This is a good section, from the results; the comparison and discussion are enough.

  • Line 375. 3.2.8. CO2 Respiration Rate

This section has a large introduction. However, this section is very complete. This section should be used as the basis for incomplete sections.

The authors should add the error bar to Figure 8, and later, the authors need to add some commentaries.

  • Line 412. 3.3.7. Mechanical properties

The graphics in Figure 9 should be identified with letters, and thus, it must be mentioned in the text (Figure 9a).

The authors should add a description of the results, since the comparison of results with other studies and discussions are good.

The Figure 9 (all the graphics) presents the same problems (error bars and statistics) as mentioned in the Figure 3.

  • Line 451. 3.3.8. Color parameters

Excellent discussions. I recommend adding some comparison with other studies.

The Figure 10 (all the graphics) presents the same problems (error bars and statistics) as mentioned in the Figure 3.

Why did the authors determine ∆E? The authors have L*. a* and b* from the control and samples.

  • Line 486. 3.4. Microbiological analysis

The graphics in Figure 11 should be identified with letters, and thus, it must be mentioned in the text (Figure 11a).

The Figure 11 (all the graphics) presents the same problems (error bars and statistics) as mentioned in the Figure 3.

The authors made a good description of results. However, the comparison with other studies and discussions are poor.

  • Line 518. 3.5. Sensory analysis

This section has a large introduction, which could well be used as a discussion.

This section only have description of results.

  1. CONCLUSIONS

The conclusions are concise and precise phrases from the results and discussions. However, this section needs to be improved from the reviewer´s observation.

REFERENCES

  • The most references have a similar format to the Author's Guide of “molecules”. However, the references 5, 7, 9, 16, 17, 31, 39, 47, 49, 58, 73, 78, 83, 89 must be rewritten, since the references have some problems in the format.

  • Why some references have doi and others references don't have doi? The authors must follow the format established by the Author's Guide of “molecules”

Author Response

GENERAL COMMENTS

Affiliations have certain errors:

- Where are the email and the initials of the authors?

R// We appreciate the reviewer's comment. The information was included.

1   Faculty of Bioscience and Technology for Food, Agriculture and Environment, University of Teramo, Via R. Balzarini 1, 64100 Teramo, Italy; [email protected] (Y.P.R); [email protected] (C.Ch. L.)

2   Facultad de Ingeniería, Programa de Ingeniería Agroindustrial, Universidad del Atlántico, Carrera 30 Número 8-49, 081008, Puerto Colombia, Colombia; [email protected] (A.S.M.)

3   Grupo de Investigación de fotoquímica y fotobiología, Universidad del Atlántico, Carrera 30 Número 8-49, 081008, Puerto Colombia, Colombia; [email protected] (C.D.G.T.); [email protected] (M.G.L); [email protected] (Y.M.P)

4   Grupo de Investigación Biotecnología, Facultad de Ingeniería, Universidad de San Buenaventura Cali, Carrera 122 # 6-65, 76001 Cali, Colombia; [email protected] (J.D.); [email protected] (D.P.N.P.)

*  Correspondence: [email protected] (C.D.G.T.) 

In all the manuscript, all ordinal numbers should be deleted and left only as a normal number.

R// We appreciate the reviewer's comment. The ordinal numbers were deleted from the manuscript

  1. INTRODUCTION

General comments

The introduction is good, since it starts from general to particular. The English is good. The objective of the study should more clearly. However, it is possible to denote some little problems in the order of some lines.  My observations:

- Line 40. What is the percentage of loss approximately?

- Line 43. In terms of tons, how much are "four thousand boxes"?

- Line 43. It is not necessary to place the reference [3] twice.

- The authors should restructure the introduction, since Lines 41-45 should be the beginning of the introduction, and then, start with "Several food losses ..."

- Lines 46-49 are similar to Lines 38-40. The authors must rearrange these lines.

Line 54. "... cultivar, microbial attacks, ..." The line should be changed to "... cultivar, and microbial attacks, ..."

Lines 49-50. "Filamentous fungi 'growth in climacteric fruits, like pears, is a significant quality problem ..."

The authors should rewrite the line, since it creates confusion for the reader.

 Lines 74-79. What is the intention with the information? These lines have no connection with the aforementioned.

Lines 99-108. What is the intention with the information? These lines have no connection with the aforementioned.

R// We appreciate the reviewer's comment. The introduction was adjusted following the suggestions (lines 38-80).  

Pears (Pyrus communis L.) is one fruit with high production figures in Latin America, with around one million tons annually [1]. China is the major producer in Asia with about 56 million tons, while Italy does the same in Europe, followed by the United States, Spain, and Turkey [2]. The immense worldwide production of pears also brings significant losses, with farmers and producers as the main affected supply chain sectors. Pear shelf-life depends on several factors such as growing season climate, nutrition, environmental conditions, cultivar, and microbial attacks, causing losses in the pear industry can be range from 5% and 50% of the production [2,3]. The primary post-harvest diseases in pear, such as blue mold, pink mold, grey mold, and black spot, are related mainly to fungi of the genus Mucor piriformis, Penicillium expansum, and Botrytis cinerea [4].

Pear fruits are climacteric, and the coating is considered one of the most popular techniques to prolong shelf life [5]. Several studies have evaluated the effect of edible coatings containing polysaccharides, such as soy protein isolate (SPI), in combination with hydroxypropyl methylcellulose (HPMC) and olive oil on 'Babughosha' Pears (Pyrus communis L.) stored at ambient temperature (28 ± 5 °C and 60 ± 10% RH). Various edible coatings like zein-oleic acid [6], plant oils [7], Shellac, Semperfresh, carboxymethylcellulose [8] are useful on pears.

Chitosan, a deacetylated derivative of chitin, is deeply investigated for many industrial and medical applications [9]. Chitosan coatings demonstrated an improvement in the storability of several perishable fruits. Barrier properties against moisture and oxygen and a capacity to preserve volatile bioactive compounds as well as a potent antimicrobial and food quality preservation capacity have been observed [10]. Some authors have evaluated chitosan coatings on pears obtaining a significantly delayed ripening and improved fruit firmness [11,12]. Nonetheless, chitosan coatings present a disadvantage its hydrophilic nature forces the introduction of hydrophobic composites such as some essential oils, which also provide antioxidant and antimicrobial properties to the food during the post-harvest stage [13].

On the other hand, the treatment of crops with essential oils (EO) could be a safer and alternative option for post-harvest preservation. Essential oils (EO) are complex mixtures of terpenes and terpenoids, coumarins, and homologs of phenylpropanoids, with biological properties and sometimes strong odors. Ruta graveolens plant has been identified as a source of different antifungal compounds such as 5- and 8-methoxy psoralen, with potent activity against fungi like  Rhizoctonia solanii, Fusarium spp., Pyrenochaeta lycopersici, Trichoderma viride, and Penicillium spp. [14]. In previous studies, the RGEO has shown high activity against the fungi Aspergillus fumigatus, Cladosporium herbarum, Candida albicans, and Colletotrichum gloeosporioides [15,16]. However, despite its antimicrobial activity, EOs are volatile and could influence the food's organoleptic properties with the possible presence of phytotoxicity [17]. A more recent and useful strategy uses composites of biopolymers and essential oils as coatings on the fruit surfaces for a controlled release of the volatile compounds inhibiting microbial growth on the fruit surface and conserving for more time nutritional composition, quality, and product acceptance [18].

Despite all the information above, no literature using CS+RGEO coatings to preserve pears' post-harvest decay is available. Taking into account the antifungal activity of RGEO components and the efficiency of CS+RGEO maintaining guava [19], cape gooseberries [20], papaya [16], and tomato [21] quality properties while inhibiting microbial growth, the objective of this study was to evaluate the effect of the application of CS+RGEO coatings on Packham's Triumph pears to preserve their quality during cold storage.

  1. MATERIALS AND METHODS

General comments

This part is complete and very clear. However, I have some observations:

  • In the subsections are not necessary to repeat the same reference.

 R// We appreciate the reviewer's comment. The adjustment was realized.

  • Line 122. 2.2 Preparation of edible coatings. 

The authors must indicate the amounts of material used, since "a specific amount" is insufficient to reproduce the experiments.

R// We appreciate the reviewer's comment. The final chitosan concentration was included allowing the reader to calculate the chitosan required for a specific emulsion volume.

 "followed our previous reported methodology [39]". The authors must rewrite these words.

"(Krauter, Colombia- previously characterized [39]),". The authors must delete these words.

 R// We appreciate the reviewer's comment. The adjustments were realized.

  • Line 130. "2.1" Coating of the fruits.

This subsection corresponds to "2.3". Thus, all the following subsections should be fixed.

"of 90 pears each".

The authors should delete the word "each".

  R// We appreciate the reviewer's comment. The adjustments were realized.

"physical-chemical and mechanical properties occurred at days 0, 3rd, 7th, 10th, 14th, and 21st. Microbiological analysis occurred during days 0, 3rd, 6th, 9th, and 12th. Sensorial properties were measured at days 0, 5th, and 10th. ".

The authors should remove ordinal numbers. The authors should leave only numbers.

 R// We appreciate the reviewer's comment. The adjustments were realized.

Why did the authors perform the physical-chemical and mechanical properties, microbiological analysis and sensory properties at different days? This is worrying, since between day 6 and day 7 (for example) multiple changes can occur, and it would affect the integrity of the samples, and thus, it would change all results obtained before and after.

 R// We appreciate the reviewer's comment. The experiment's idea was to validate the coatings' effect on the post-harvest quality of pears due to the characteristics necessary for the product to be suitable for sale and accepted by the buyer. A time window of 12 days was used for the microbiological study (with a data collection every three days), an adequate time to visualize the fruit's behavior, since more extended than that, the microbiological population is too high. The sensory analysis was developed until day ten because, after this time, the severity index of the control fruit is high and not suitable for consumption by the assessors.

The study of the physicochemical and mechanical properties was carried out for a longer time to demonstrate the fruit's emulsions' impact until the deterioration of the fruit and because the fruits were not mechanically or physically deteriorated.

  • Line 155. 2.3 Post-harvest quality properties of Pears

The authors do not need to add subsections, since the authors used the same methodology [39].

R// We appreciate the reviewer's comment, but under our criteria, the subsections bring order to the work.

The authors must write the characteristics of all equipment used (model, country, etc.).

 R// We appreciate the reviewer's comment. The information was included.

  • Line 182. 2.4. Decay evaluation

The authors should remove ordinal numbers. The authors should leave only numbers.

  R// We appreciate the reviewer's comment. The adjustments were realized.

  • Line 209. 5. Sensory Activity

The authors should indicate more details regarding the average age, number of women, number of men, etc.

 R// We appreciate the reviewer's comment. The following information was included (lines 185-192):

The test was carried out with one group of 50 non-trained assessors made-up women and men in equal numbers with an age range between 19 and 26 years during days 0, 5, and 10 of storage [21]. All the tests were performed in the morning hours. The samples were cut into cubes of the same size. Then, the peel was not removed by the pear consumers' preferences in Colombia. The assessors were informed of the methodology and signed an informed consent with the test details. Attributes as color, flavor, aroma, texture, and gloss were evaluated in the test. Assessors were asked to score the difference between samples where 0–2 represented extreme dislike; 3–5 fair; 6–8 good; and nine excellent for each attribute.

  1. RESULTS AND DISCUSSION

General comments

The results and discussion section should contain a description of the results, comparison with other studies and discussion of the results obtained with respect to other studies.

The authors should add the entire error bar in all the figures.

All subsections have only descriptions of results. Therefore, the authors should delve into comparison of results and discussions. The introduction that begins each subsection may help to the authors as a discussion.

R// We appreciate all the reviewer's comments. As can be seen, we added more discussion and comparison of the results with other studies, and the entire error bar was added in all the graphs.

  • Lines 223-227. All values must be inserted in the supplementary section material.

  R// We appreciate the reviewer's comment. The information was included as supplementary material.

  • Lines 229-231. These lines should be deleted, since all this was explained in the section materials and methodology.

   R// We appreciate the reviewer's comment. The adjustments were realized.

  • Line 232. 3.2.1. pH analysis

The Figures 3 shows problems about the statistics. For example, at day 21, the samples have different letters (F1 = b, F2 = a, F3 = ab, F4 = ab and F5 = b) and the authors indicate "not present significantly different".

Thus, why the letters are different?. Additionally, the comparison between days has the same problem, since, for example, at day 14 and day 21 there are no significant differences, but the authors mention it with different letters.

R// We appreciate the reviewer's comment. However, it had been a misunderstanding when the statistical analysis was reviewed for all figures. The mentioned letters correspond to the significant differences in each treatment concerning the factor time. The statistical analysis to determine the differences between control and formulations was the Dunnet test. It has changed by the Duncan test to determine significant differences (uppercase letters) with control and between them. The information was included in the statistical analysis section and figure heading (lines 194-199):

A completely randomized design was applied in this study. All experiments were conducted in triplicates except for CO2 respiration rate (n = 2). The data from each sampling point are shown as the mean ± SD and were statistically evaluated by ANOVA, followed by individual comparisons using Duncan's Multiple Range Test, with a confidence level of 95% (α = 0.05). This test was used to assess the effect of diverse formulations on the response variables described previously for each time point. The Fisher test was used to assess the differences between days for each treatment.

The results and discussion were changed according to the new statistical analysis (lines 208-221):

"All the pH formulations do not present significant differences concerning the control or between them except for days 10 and 21 (Figure 3). These days, the formulations added a protective effect and retained the organic acid content in the fruit; however, no effect is seen from the inclusion of the RGEO to the CS formulation. The statistical analysis for each formulation relating to storage days showed no significant difference between the first day and the final day for the formulations confirming coatings' protective trend. Uncoated fruits presented significant differences concerning the factor days caused by fast organic acids' consumption. The present study results are also in agreement with previous reports that shown similar trends in pears with chitosan-based coatings storage at 20 °C with pH also ranges between 4.2 and 4.6 [24]. The ripening processes demand high energy from different carbon compounds (e.g., organic acids, amino acids, and sugars) in the metabolisms pathways [25]. The pH usually increases during the ripening of a climacteric fruit due to the organic acids' consumption for the metabolic processes during fruit respiration [26]. It is noteworthy that some authors suggested a reduction in the rate metabolism of the fruit [27], as a phenomenon caused by a barrier effect of the coatings, behavior that which may explain the lower pH values in coated pears."

  • Line 248. 3.2.2. Titratable Acidity (TA)

This section has a small mention of comparison of results with other study, but it is insufficient.

The Figure 4 presents the same problems (error bars and statistics) as mentioned in the Figure 3.

 R// We appreciate the reviewer's comment. The adjustments were realized (lines 232-249).

TA can be correlated with the primary organic acids in fruits, mainly citric acid and maleic acid in pears, according to the proteomic analysis in the ripening process [28]. The formulations do not show significant differences between them or the control for the days 0, 3, 7, 10, 14 (Figure 4). On day 21, coated pears with F2, F3, and F5 tend to maintain significantly higher levels of TA than compared with the control, demonstrating protection (lower consumption of organic acid and ripening) during the final day of the storage time. The highest level of TA (0.17%) was recorded in pears coated with F5.  As in the case of pH, no effect is observed of the different concentrations of RGEO in formulation with CS. The TA was also significantly different (p<0.05) on day 21 compared to the first day to the uncoated pears and F4. The other formulations do not present significant differences (p<0.05), with the storage time, confirming the protective effect of F2, F3, and F5. Similar trends have been observed with pears coated with coating CS-based, corroborated by Rosenbloom et al.[12], and Lin et al. [29]. The change of TA with storage time in uncoated fruits may be due to the normal ripening process [30], which is slowed down with the formulations with CS and CS+RGEO by the barrier effect of the coatings against oxygen, which inhibit the oxidation of the organic acids [31]. Previous studies have demonstrated that chitosan-based coatings reduce citric and malic acid contents, which are the major organic acid in ripe pear fruit [28]. Organic acids can support numerous and diverse functions in plants associated with the supply of carbon skeletons during the plant growth process or the critical role of malate in photosynthesis and stomatal regulation [32].

  • Line 268. 3.2.3. Soluble Solids Content (SS) 

This section has a large introduction, which could well be used as a discussion.

The Figure 5 presents the same problems (error bars and statistics) as mentioned in the Figure 3.

  R// We appreciate the reviewer's comment. The adjustments were realized (lines 259-275).

To characterize better the CS-RGEO coatings effects on pears, we analyzed the content of soluble solids content. The results showed in Figure 5 indicated statistically significant differences (p<0.05) in the SS content between F5 and F4, F5 regarding uncoated pears for days 10 and 21. No effect significative in the solids soluble total was evidenced between the pears coated with the formulations indicating there is no more significant influence of the coatings with the inclusion of RGEO. However, the lowest percentage of soluble solids (11.45%) on the last day of storage was recorded in pears coated with CS + RGEO 1.5% (F5).

The uncoated pears increased significantly from 10.3% of SS to 12.8%, after 21 days of storage at 18°C, while pears coated with F5 do not present differences significant and have a minor increase of all formulations with a change from 10.4% to 11.6%, demonstrating a preservation effect previously seen with the pH and acidity titratable. These results were consistent with the result showed by Rosenbloom et al.[12] with CS-based coatings, it is observed that fruits coated have a minor change in the solids soluble during storage time. According to the maturation stage and stress conditions, the ripening process drastically altered the biochemistry and physiology of climacteric fruits such as pears due to by-products and changes produced in the secondary metabolism. Sugar accumulation (mainly glucose, fructose, and sucrose) occurs exponentially upon the cell division phase, to finally plateau at the end of the ripening process, while organic acid contents substantially decrease [33].

  • Line 291. 3.2.4. Maturity Index

This section has a small discussion, but it is insufficient.

The Figure 6 presents the same problems (error bars and statistics) as mentioned in the Figure 3.

   R// We appreciate the reviewer's comment. The adjustments were realized (lines 287-305).

The maturity index is the ratio between the percentage of soluble solids and the percentage of acidity. Figure 6 shows that all the samples increased the maturation index during the storage time but slower than the control. All the formulations kept a significantly lower maturation index than the uncoated samples (F1) for the 21st day. Despite the formulations not present significative differences between them, the maturity index lower values were obtained with F3 (RGEO 0.5%) and F5 (RGEO 1.5%) with 73 and 71%, respectively. These results showed the same trend, followed by properties already studied where all the formulations have an effect protector in the pears. The formulations presented significant differences regarding the storage time with an increase since day 7. However, the change of the value of maturity index is lower in the coated fruits. The results obtained in this study are in agreement with previous studies in guava [19], cape gooseberries [20], tomatoes [21], and papaya [16] coated with CS+RGEO. The main modifications during fruit ripening are changes in color and textural based on sugar, organic acid, and volatile compounds, contributing to the balance between sugar and organic acids [28]. The changes in the percentage of soluble solids and the percentage of acidity and, therefore, in the maturity index during storage in fruits are the ripening process results [27]. On the other hand, an increase in the solids soluble level during storage is because starch and nutrients can degrade into sugars and soluble substances [35]. The chemical composition of the RGEO, mainly composed of ketones and sesquiterpenes with the ability to modify the internal atmosphere of the fruits, also can contribute to the protective effect is CS coatings-based [19].

  • Line 317. 3.2.5. Weight Loss

This section has a small discussion, but it is insufficient.

The Table 1 presents a statistical problem. Example, in CS+0%RGEO, Day 7 presents "ab", but the result has significant different with the other days.

R// We appreciate the reviewer's comment. The adjustments were realized (lines 318-334).

Weight loss percentage is an essential indicator of the degree of preservation of fruits and vegetables. As observed in Table 1, F2 no significant differences with the control on no day of the study. F5 and F3, F4, F5 showed significant (p<0.05) lower weight losses than the control, for day 14 and 21, respectively, according to the Duncan test. On day 21, there are no significant differences between CS and CS+ RGEO formulations. However, a protective effect of the RGEO is evident with weight losses less than control in a range between 41- 50%. All the samples showed significant (p<0.05) weight losses with time according to the LSD Fischer test by day 21 concerning day 3. Several authors have also reported the same trend with a reduction in weight loss rate in the fruits with CS coatings-based reinforced with essential oils. Peralta et al. [16] reported weight losses of 20% in papayas coated with CS-RGEO formulations after 12 days of storage, evidencing a good behavior of the coatings in this study. Our results in this study evidence a good behavior protector of the coatings with similar values but more extended storage. Cosme Silva et al. [36] suggest that the CS coatings-based reduce the fruits' respiratory rate acting as a barrier and decreasing water loss and solutes.  Usually, the weight loss corresponds to the unbonded water released to the environment [22]. Our results agree with those reported by various authors who suggest that the inclusion of the hydrophobic compounds in the CS formulations provides a better water loss barrier [13]. The RGEO reinforced the coatings and decreased weight losses in the pears.

  • Line 331. 3.2.6. Severity Index (SI)

What is DI?

This section presents a good comparison with other studies, but a little discussion with the results of the authors.

The Figure 7 presents the same problems (error bars and statistics) as mentioned in the Figure 3.

R// We appreciate the reviewer's comment. The adjustments were realized (lines 470-481).

All the formulations presented a significant decrease in the SI concerning uncoated pears (Figure 10). On day 21, is observed significant differences between CS+RGEO and CS formulation. F4 and F5 (CS+RGEO 1.0% and CS+RGEO 1.5%) preserved the pears until day 21st without any physical or microbiological damage. Uncoated samples ranged from 0 on day 0 to 4.0 on day 21 of the storage. The formulation containing CS+RGEO 0.5% presented a protective effect, with only a 0.7% of SI on day 21st. Formulation F2 also presented a protecting effect with 1.7% of SI on day 21st. Figure 7 shows that only the control has significant differences for SI according to the LSD Fisher Test regarding factor time, evidencing the positive effect of the formulations in the conservation of pears. Those results demonstrated that CS presented a protective effect against physical and microbiological damage, acting as a positively reinforced barrier with the addition of RGEO in a concentration-dependent manner.

  • Line 358. 3.2.7. Disease damage incidence and Infection index

This is a good section, from the results; the comparison and discussion are enough.

  • Line 375. 3.2.8. CO2 Respiration Rate

This section has a large introduction. However, this section is very complete. This section should be used as the basis for incomplete sections.

The authors should add the error bar to Figure 8, and later, the authors need to add some commentaries.

 R// We appreciate the reviewer's comment. The adjustments were realized (lines 363-374).

Figure 7 shows the high respiration rate of uncoated fruits at the beginning of the study compared to coated fruits presenting significant differences. After three and seven days, there was an apparent decrease in the respiration rate due to the treatments. On day 14, no significant differences were observed between treatments. On the last day, F3 has a significant decrease concerning the control and the other formulations. Besides, it showed significant differences regarding factor time in day 21 compared to day 0. F4 and F5 had a higher respiration rate of the treatments.

Chitosan and essential oil coatings have been previously documented as suitable treatments with the capacity to inhibit O2 consumption and CO2 production, generating lesser gas permeability during the first ten storage days. After that, the barrier effect is lost as a result of the deterioration of the coatings. Coating hydration and oil evaporation could explain the decreasing barrier effect, and the fruit will continue in its climacteric peak due to senescence [19].

  • Line 412. 3.3.7. Mechanical properties

The graphics in Figure 9 should be identified with letters, and thus, it must be mentioned in the text (Figure 9a).

 The authors should add a description of the results, since the comparison of results with other studies and discussions are good.

The Figure 9 (all the graphics) presents the same problems (error bars and statistics) as mentioned in the Figure 3.

 R// We appreciate the reviewer's comment. The adjustments were realized (lines 392-410).

Regarding compression resistance (Figure 8a), on day 14, F3 presents significant differences concerning the control. For the other days, the formulations do not present significant differences between them or with the control. Besides, no significant differences were found between the force (N) (Figure 8b).  Likewise, it is observed that concerning deformation (Figure 8c), no differences significative between formulations neither with control from day 3. Despite not having statistically significant differences in the mechanical properties of the fruits coated with the different formulations, the texture properties of the pears were conserved for all treatments. Similar results were reported by Dai et al. [47], who evaluated coatings based on starch and starch nanocrystals on pears. In this study, even though there were no significant differences between the treatments, they did occur between coated and uncoated pears.

The formulation that provides the most mechanical resistance to the pear is CS+RGEO 0.5% (F3), which presents the highest value after 21 days of storage, both for compression resistance (7.1 kPa) and force (20.1 N). That the coating with less RGEO present better mechanical properties values may be related to saturation. This phenomenon indicates a limiting concentration of essential oil above which a decrease in mechanical properties is promoted; the excess increases intermolecular mobility, favoring the solubilization of components and forming different bonds that could reduce the coating's cohesion, preventing it from fulfilling its barrier function [48]. F3 presented the lower value of weight loss on day 21.

  • Line 451. 3.3.8. Color parameters

Excellent discussions. I recommend adding some comparison with other studies.

The Figure 10 (all the graphics) presents the same problems (error bars and statistics) as mentioned in the Figure 3.

R// We appreciate the reviewer's comment. The adjustments were realized (lines 431-435 and 452-457).

When adding the coatings on day zero, no significant differences were observed between the treatments and the control for L and b*, only a present difference significative for a* but with a minimal change, which indicates that the coatings did not alter the original color of the pears, presenting the transparency expected of an edible coating. Peralta et al. [16] also reported no significant change in color parameters between uncoated papaya and coated fruits.

The samples with RGEO significantly decreased b* concerning the uncoated pears on day 21. That result suggests that the observed maturation process's decrease may be related to protecting the polyphenolic compounds present on the pear's surface by the coating [55]. Similar results were previously obtained in other researches with chitosan-essential oil coatings in papaya [16], guava [19], and strawberries [56], where the changes in color parameters are delayed in coated fruits regarding uncoated fruit.

Why did the authors determine ∆E? The authors have L*. a* and b* from the control and samples.

 R// We appreciate the reviewer's comment. According to the articles consulted related to pears in the scientific literature, we found that it is enough to characterize the fruits' color change with the determined parameters to analyze the coatings' effects on the fruit's color.

  • Line 486. 3.4. Microbiological analysis

The graphics in Figure 11 should be identified with letters, and thus, it must be mentioned in the text (Figure 11a).

The Figure 11 (all the graphics) presents the same problems (error bars and statistics) as mentioned in the Figure 3.

The authors made a good description of results. However, the comparison with other studies and discussions are poor.

 R// We appreciate the reviewer's comment. The adjustments were realized (lines 516-532 and 542-551).

During day 12th of storage treatment, coatings F2 (CS) and F3 (CS+RGEO 0.5%) inhibited the bacteria growth for about 0.8 and 2.7 Log CFU/g after treatment. Coatings with 1.0% and 1.5% of RGEO (F4 and F5) showed more potent inhibition of about 2.8 and 3 Log CFU/g, respectively; no significant differences were evidenced between CS+RGEO formulations demonstrating there is no dependence on the essential oil concentration. As shown in Figure 11, there were no significant differences (p<0.05) between the 6th and 12th day to F3 and F4, also present similar inhibition than F5, which is advantageous since a lower amount of RGEO could be used in the formulation with an excellent mesophilic bacteria inhibition during more the storage time. Besides, F3 and F4 no present significant differences in respect factor time from day 6, suggesting that one of these two formulations could be the most suitable from the microbiological point of view. Similar results were reported in tomatoes [21] and guava [19] coated with CS+RGEO. Usually, the mesophilic bacteria count describes the population of coccus, bacillus, and spiral bacteria present in the fruit; when the levels are high, this could indicate poor hygienic conditions and decay of fruits [21]. Many studies have reported the antimicrobial effect of CS against Gram‐positive and Gram‐negative bacteria [18], which can be reinforced with the antimicrobial activity present in the essential oils; it has been reported efficacy of RGEO against bacteria like Escherichia coli, Bacillus cereus, Pseudomonas aeruginosa, and Staphylococcus aureus [16].

The statistical analysis indicated a significant difference (p<0.05) during the storage time for F1 (Figure 11 B). However, F2 is statistically equal between days 3 and 6 and the 9 and 12. From day 3, the formulations present significant differences concerning control; on day 9, F4 has significant differences regarding F2, showing the RGEO inclusion effect. On the last day of storage time, pears coated with F2 (CS) only presented a reduction of 1.2 Log CFU/g on day 12th, F3 (CS+RGEO 0.5%), and F4 (CS+RGEO 1.0%) demonstrated 2.4 and 2.55 Log CFU/g reductions, respectively. F5 (CS+RGEO 1.5%) presented a reduction of 3.3 Log CFU/g in molds' growth, indicating a potent antifungal effect dependent on the RGEO amount in the coating. Previous studies with CS+RGEO coatings in tomatoes reported a reduction of 2 Log CFU/g in molds' growth after 12 days of storage with RGEO 0.5% [21]. Xu et al. [59] reported the better inhibition in the mold's growth in pear with chitosan-coatings reinforced with cinnamon oil compared to that obtained with just chitosan.

  • Line 518. 3.5. Sensory analysis

This section has a large introduction, which could well be used as a discussion.

This section only have description of results.

 R// We appreciate the reviewer's comment. The adjustments were realized (lines 580-590).

Palatability is one of the most critical organoleptic properties of foods, and it is mainly dependent on the balance between organic acids (acidity) and sugar (sweetness), indicating the importance of the metabolic pathway of the fruit for its value [33]. One of the biggest problems with the use of essential oils in edible coatings is the transfer of compounds from oil to the fruit that can change the product's palatability [16]. As previously evidenced, the CS-RGEO coatings delayed the pears' ripening; this could influence the coated samples' acceptance compared to the uncoated ones. One of the biggest problems with using essential oils in edible coatings is transferring compounds from oil to the fruit that can change the product's palatability. In our study, CS-RGEO coatings did not show flavor transfers according to the sensory assessors' opinions. Similar results were obtained in papaya [16], guava [19], and fresh-cut pears [24], where fruit treated with chitosan-based coating reinforced with essentials oils or natural extracts received better acceptance than uncoated fruit.

  1. CONCLUSIONS

The conclusions are concise and precise phrases from the results and discussions. However, this section needs to be improved from the reviewer's observation.

R// We appreciate the reviewer's comment. The adjustments were realized according to the changes made (lines 592-610).

In the present work, we demonstrated a protective effect in the quality properties of pears after surface application of chitosan-Ruta graveolens (CS+RGEO) essential oil coatings in four different concentrations (0%, 0.5 %, 1.0 %, and 1.5 %, v/v) during 21 days of storage under 18°C. The fruits' physical-chemical characteristics were analyzed, evidencing a protecting effect of coatings against the ripening process; despite no significant differences between CS and CS-RGEO formulations, there is no affectation on the properties evaluated. Mature index, decay index, disease damage incidence, and color results correspond to less ripped fruits. A weight-loss reduction of 50% (from 40.2±5.3 to 20.3±3.9) compared to the uncoated pears was evident with CS+RGEO 0.5%, demonstrating a barrier effect of the coatings. After mechanical properties analysis for the coated and uncoated fruits, the pear's most mechanical resistance was obtained with CS+RGEO 0.5% after 21 days of storage, both for compression resistance (7.42 kPa) and force (22.7 N).

McKinney index indicated total protection for pears coated with CS+RGEO1.5%. Formulations including 15 L/mL of RGEO significantly reduced in 3.0 Log CFU/g aerobic mesophilic bacteria and in 3.3 Log CFU/g molds and yeast compared to control without affecting consumer perception.

The sensorial analysis of the coated fruits demonstrated that all the formulations were acceptable for the organoleptic attributes and were able for human consumption. The study showed that the formulations including CS+RGEO 0.5% are suitable for post-harvest treatment for pears and showing adequate antimicrobial protection, with a lower oil concentration (that would be convenient in economic terms), improving consumers' acceptance 

REFERENCES

  • The most references have a similar format to the Author's Guide of "molecules". However, the references 5, 7, 9, 16, 17, 31, 39, 47, 49, 58, 73, 78, 83, 89 must be rewritten, since the references have some problems in the format.
  • Why some references have doi and others references don't have doi? The authors must follow the format established by the Author's Guide of "molecules"

 R// We appreciate the reviewer's comment. The adjustments were realized.

Round 2

Reviewer 2 Report

Dear Author(s)

After an exhaustive revision, the manuscript is Accept in present form. The resubmitted manuscript has been completely improved compared to its previous version. Therefore, the manuscript can be published in “Molecules”.

Best regards
